# RORγ is a targetable master regulator of cholesterol biosynthesis in a cancer subtype

Demin Cai [1], Junjian Wang[1], Bei Gao[2], Jin Li[1], Feng Wu[3], June X. Zou[1], Jianzhen Xu[4], Yuqian Jiang[1], Hongye Zou[1], Zenghong Huang [1], Alexander D. Borowsky[5], Richard J. Bold[6,7], Primo N. Lara[7], Jian Jian Li [8], Xinbin Chen [7,9], Kit S. Lam[1,7], Ka-Fai To[3], Hsing-Jien Kung[1,7], Oliver Fiehn [2], Ruqian Zhao[10,11], Ronald M. Evans[12] & Hong-Wu Chen[1,7]*

Tumor subtype-specific metabolic reprogrammers could serve as targets of therapeutic intervention. Here we show that triple-negative breast cancer (TNBC) exhibits a hyper-activated cholesterol-biosynthesis program that is strongly linked to nuclear receptor RORγ, compared to estrogen receptor-positive breast cancer. Genetic and pharmacological inhibition of RORγ reduces tumor cholesterol content and synthesis rate while preserving host cholesterol homeostasis. We demonstrate that RORγ functions as an essential activator of the entire cholesterol-biosynthesis program, dominating SREBP2 via its binding to cholesterol-biosynthesis genes and its facilitation of the recruitment of SREBP2. RORγ inhibition disrupts its association with SREBP2 and reduces chromatin acetylation at cholesterol-biosynthesis gene loci. RORγ antagonists cause tumor regression in patient-derived xenografts and immune-intact models. Their combination with cholesterol-lowering statins elicits superior anti-tumor synergy selectively in TNBC. Together, our study uncovers a master regulator of the cholesterol-biosynthesis program and an attractive target for TNBC.

[1] Department of Biochemistry and Molecular Medicine, University of California Davis, Sacramento, CA, USA. [2] West Coast Metabolomics Center, University of California Davis, Davis, CA, USA. [3] Department of Anatomical and Cellular Pathology, The Chinese University of Hong Kong, Hong Kong SAR, China. [4] Shantou University Medical College, Shantou, China. [5] Department of Pathology and Laboratory Medicine, University of California Davis, Sacramento, CA, USA. [6] Department of Surgery, University of California Davis, Sacramento, CA, USA. [7] Comprehensive Cancer Center, University of California Davis, Sacramento, CA, USA. [8] Department of Radiation Oncology, University of California Davis, Sacramento, CA, USA. [9] Comparative Oncology Laboratory, University of California Davis, Davis, CA, USA. [10] MOE Joint International Research Laboratory of Animal Health and Food Safety, Nanjing Agricultural University, Nanjing, China. [11] Key Laboratory of Animal Physiology and Biochemistry, College of Veterinary Medicine, Nanjing Agricultural University, Nanjing, China. [12] Gene Expression Laboratory, Salk Institute, Howard Hughes Medical Institute, Salk Institute, La Jolla, CA, USA. *email: hwzchen@ucdavis.edu

The mevalonate–cholesterol biosynthesis pathway produces sterols, isoprenoids, and ubiquinone that are essential for tumor growth. Indeed, deregulated lipid and cholesterol homeostasis are often associated with tumorigenesis and cancer progression[1]. Elevated expression of the cholesterol-biosynthesis pathway is strongly associated with poor prognosis in the majority of solid tumors including breast cancer. Intra-tumor accumulation of cholesterol esters and metabolites is correlated with tumor cell hyper-proliferation, metastasis, stemness, and therapeutic resistance[2–7]. Deregulated cholesterol-biosynthesis flux or expression of the pathway enzymes could be tumor driving[3,8,9]. However, the mechanisms underlying the deregulation are poorly understood.

The cholesterol-biosynthesis pathway is under tight regulation by transcription factors, such as sterol regulatory element-binding protein 1 and 2 (SREBP1 and 2) and nuclear receptor (NR) liver X receptors (LXRs)[1,10,11]. SREBPs, particularly SREBP2, play a predominant role in the control of cholesterol biosynthesis. In response to low cholesterol levels in the endoplasmic reticulum, SREBP2 is cleaved by proteases. The cleaved, N-terminal portion of SREBP2 translocates to the nucleus and binds to sterol response elements (SREs) to activate the expression of cholesterol-biosynthesis enzymes, such as HMGCR and SQLE. As membrane cholesterol level increases, less SREBP2 is activated. This SREBP2-mediated regulatory circuit is believed to be hard-wired in cholesterol-producing normal tissues and is widely perceived to function similarly in tumors of most cancer types. In addition, SREBPs have also been shown to be the downstream targets and effectors of oncogenic signaling, such as the pRb, Myc, PI3K-AKT, or mTORC1 pathways[1]. Intriguingly, gain-of-function mutant forms of p53 display oncogenic properties partly through upregulation of SREBP2 targets[12]. However, little is understood at the chromatin level how SREBP2 function is regulated in its activation of the program.

Triple-negative breast cancer (TNBC) tumors feature a high proliferative index compared with estrogen receptor-α-positive (ER+) tumors and still lack effective targeted therapy[13]. Metabolic reprogramming in breast cancer, particularly in lipid and cholesterol pathways, is still poorly defined and remains to be exploited for therapeutic targeting[14]. Recent studies suggest that circulating cholesterol-lowering drug statins can increase expression of cholesterol-biosynthesis pathway genes/enzymes in tumors, likely due to a heightened feedback regulation mediated by SREBP2[1,9,15–18]. Therefore, strategies to suppress tumor hyperactive cholesterol-biosynthesis program without induction of the feedback response are desirable for effective intervention. In this regard, small molecules, such as fatostatin, that target SREBP proteins have shown cancer cell-killing effects without the feedback[19]. However, their apparent toxicity limits their clinical use. One major family of lipid metabolic regulators is NRs. Indeed, promoting tumor cholesterol efflux by a LXR agonist has been shown effective in treating subtypes of glioblastoma[5], raising the possibilities of other NR involvement.

RAR-related orphan receptor gamma (RORγ), together with related RORα and RORβ, constitutes a subfamily of the NR superfamily of transcription factors that are attractive therapeutic targets for metabolic and autoimmune diseases[20]. RORγ as well as RORα play important roles in the control of hepatic circadian rhythmic expression of glucose and lipid metabolic genes[21]. By interrogating the distinctions of TNBC in metabolic reprogramming, here we identify RORγ as an essential driver of the cholesterol-biosynthesis program in TNBC. RORγ inhibition negates statin-induced, SREBP2-dependent feedback regulation, and decreases tumor cholesterol-biosynthesis rate without affecting host cholesterol homeostasis. Therefore, our findings define RORγ as a previously unsuspected master regulator of

cholesterol biosynthesis in cancer metabolism and as an attractive therapeutic target.

## Results

**A hyper-activated cholesterol biosynthesis is linked to RORγ.** Tumor gene expression profile analyses have yielded valuable information for TNBC subtyping and therapeutic targeting, despite its high heterogeneity. We thus interrogated the METABRIC dataset (1459 ER+ and 313 TNBC) for major metabolic pathways that are distinctly altered in TNBC when compared with ER+ tumors. The analysis showed that the expression of cell cycle/proliferation genes was elevated more in TNBC than in ER+ tumors, and that a subset of TNBCs displayed aberration in fatty acid oxidation or metabolism (Fig. 1a, Supplementary Fig. 1a). Notably, one major distinction between the two tumor types was that the vast majority of TNBCs displayed a highly elevated cholesterol-biosynthesis program. In support of the distinction of TNBC possessing a hyper-activated cholesterol-biosynthesis program, our metabolomics analysis of a cohort of tumor tissues (218 ER+ and 33 TNBC) demonstrates that TNBC tumors had significantly higher cholesterol contents than those of ER+ ones (Fig. 1b).

To identify potential driver(s) of the aberrant cholesterol biosynthesis in TNBC, we treated TNBC cell MDA-MB468 and ER+ cell MCF7 with a panel of 31 small molecules targeting the NR family members and 2 compounds that target SREBP2 translocation regulators SCAP (fatostatin) or S1P (PF-429242) (Fig. 1c, Supplementary Fig. 1b). As expected, treatment with fatostatin or PF-429242 strongly decreased the expression of key cholesterol-biosynthesis genes. Consistent with previous study[22], statin (atorvastatin/ATV) treatment resulted in increases of cholesterol-biosynthesis gene, due to its relief of the feedback suppression by cholesterol. As reported[5], the LXRα agonists induced cholesterol efflux gene ABCA1 in both TNBC and ER+ cells. Notably, among the NR modulators, two RORγ antagonists (XY018 and GSK805) displayed the strongest inhibition of all five cholesterol-biosynthesis genes in the TNBC cells but not in the ER+ cells. Although cholesterol precursors or metabolites have been shown to bind to RORγ and regulate its transcriptional activity particularly in Th17 cell differentiation[23–25], so far RORγ itself has not been implicated in the direct control of cholesterol homeostasis. In support of the notion that RORγ is a candidate driver of the aberrant cholesterol-biosynthesis program in TNBC, analysis of the METABRIC dataset revealed that the expression of RORγ gene RORC, but not the other NRs, had a significant positive association with cholesterol-biosynthesis program in TNBC, not in ER+ tumors (Fig. 1d, Supplementary Fig. 1c–f). Moreover, expression of RORC, not SREBP2, is linked with poor overall survival selectively in TNBC (Supplementary Fig. 1g). Together, these data suggest that cholesterol-biosynthesis program is elevated distinctively in TNBC, and that the NR member RORγ is a candidate driver of the cholesterol-biosynthesis aberration.

**RORγ is a major driver of TNBC cell growth and survival.** To examine the function of RORγ in TNBC, we performed CRISPR–Cas9 knockout and siRNA knockdown of RORC in multiple TNBC and other models. As shown in Fig. 2a and Supplementary Fig. 2a, b, knockout of RORC markedly inhibited the growth of all six TNBC cell lines examined, but not that of ER+ MCF-7 or the nonmalignant MCF-10A cells. The knockout also resulted in a poor survival of TNBC cells as measured with colony formation and caused pronounced apoptosis as reflected by activation of caspase3/7 (Fig. 2b, c, Supplementary Fig. 2c). Similarly, siRNA silencing of RORC also significantly inhibited

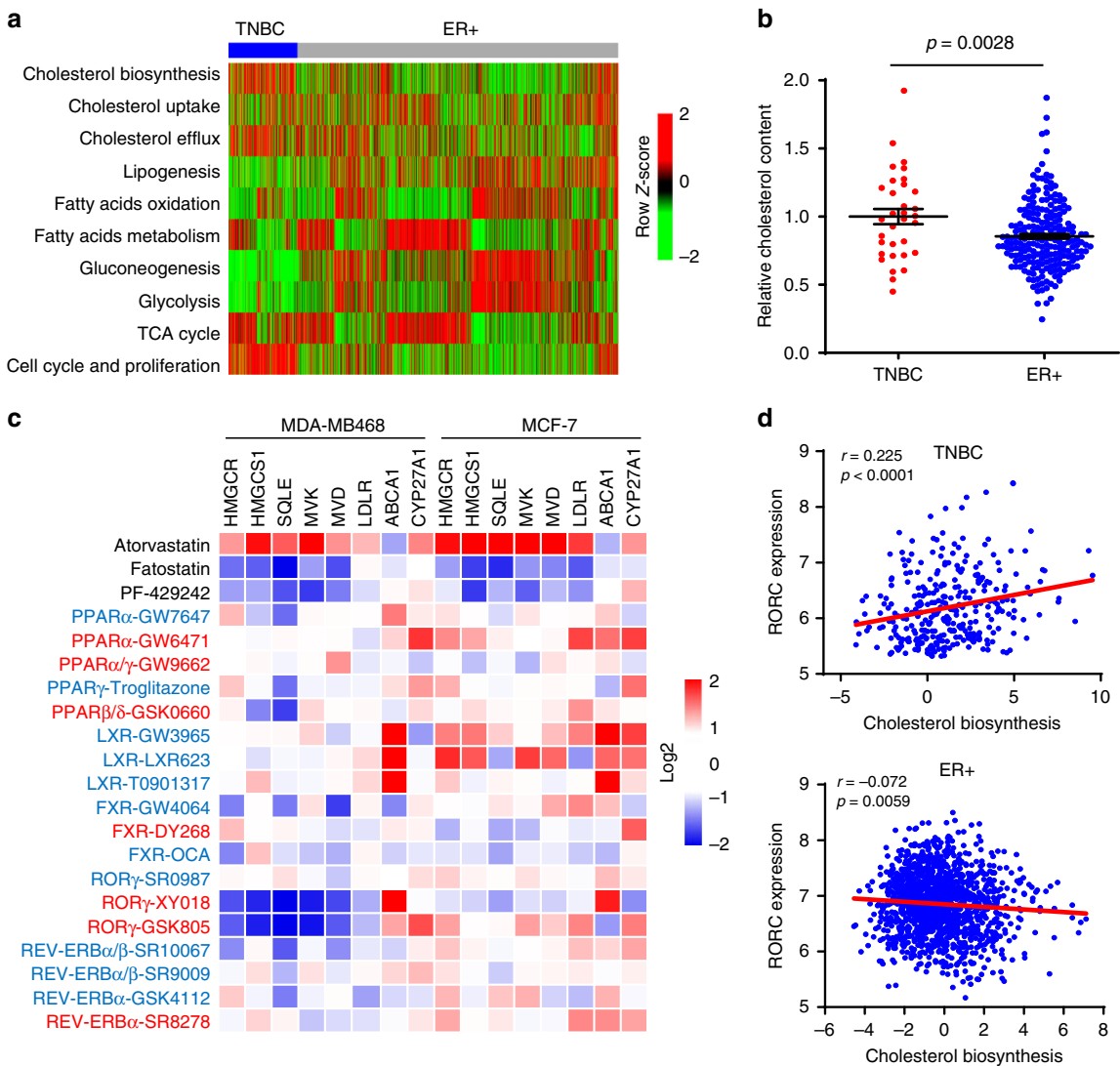

**Fig. 1** A hyper-activated cholesterol-biosynthesis program is linked to RORγ in TNBC. **a** Heat map of activity scores of gene expression of indicated pathways with data normalized from −2 (Green) to 2 (Red) in tumors of TNBC (313 patients) and ER+ cancer (1459 patients). The gene expression datasets were from METABRIC consortium. For each of the ten metabolic pathways and cell cycle/proliferation pathway, a gene set was manually compiled, and an "activity score" was represented by the first principal component of the gene set for each of the ER+ or TNBC samples. **b** Relative cholesterol contents of 33 TNBC tumors and 218 ER+ tumors were measured with GC–TOFMS. Student's t test. P = 0.0028. **c** Heat map display of fold changes (in log2) in cholesterol homeostasis gene expression in MDA-MB468 and MCF-7 cells treated for 48 h by agonists (blue) or antagonists (red) of indicated NRs. The concentration was 2.5 μM for each compound. The expression of indicated genes was analyzed by qRT-PCR. The experiments were repeated three times. **d** Scatter plots showing correlation of transcript expression between RORC and cholesterol-biosynthesis pathway in TNBC and in ER+ tumors. The Pearson correlation metric was computed by using the 'cor' function in R. The METABRIC dataset was used

the TNBC cell growth (Supplementary Fig. 2d). Moreover, RORC silencing also led to significantly decreased expression of proteins that are important for cell proliferation, growth, and survival, along with an increased level of cleaved PARP-1 (Supplementary Fig. 2e). To determine whether elevated RORγ alone is sufficient to promote the growth and survival of TNBC cells, RORγ was overexpressed in the TNBC cells. The overexpression significantly enhanced the colonogenic growth in regular growth conditions (Fig. 2d, Supplementary Fig. 2f).

Having demonstrated the crucial role of RORγ in TNBC cells, we next examined whether its antagonists possess strong growth-inhibitory effects. GSK805 was demonstrated to be effective in suppression of Th17 cell differentiation via antagonizing RORγt[26], while XY018 was shown to strongly inhibit the growth of androgen receptor-positive prostate

tumors via inhibition of tumor cell RORγ[27]. Both compounds displayed excellent selectivity toward RORγ[28]. We treated a panel of TNBC and ER+ human breast cancer cell lines and several mouse ER-negative cell lines with the two antagonists and two RORγ agonists along with the PARP-1 inhibitors (Fig. 2e, Supplementary Fig. 2g) for comparison. Consistent with the distinct function of RORγ in TNBC revealed by our genetic approach, the two RORγ antagonists displayed strong growth inhibition in most of the human TNBC cells and the highly metastatic mouse mammary tumor cells (4T1 and MET-1) (Fig. 2e). In contrast, they had only negligible or limited growth-inhibition effect on ER+ cells. However, the protein expression of RORγ in ER+ breast cancer cells is no less than most of the TNBC cells (Supplementary Fig. 2h). Similar selectively inhibitory effects of the antagonists on

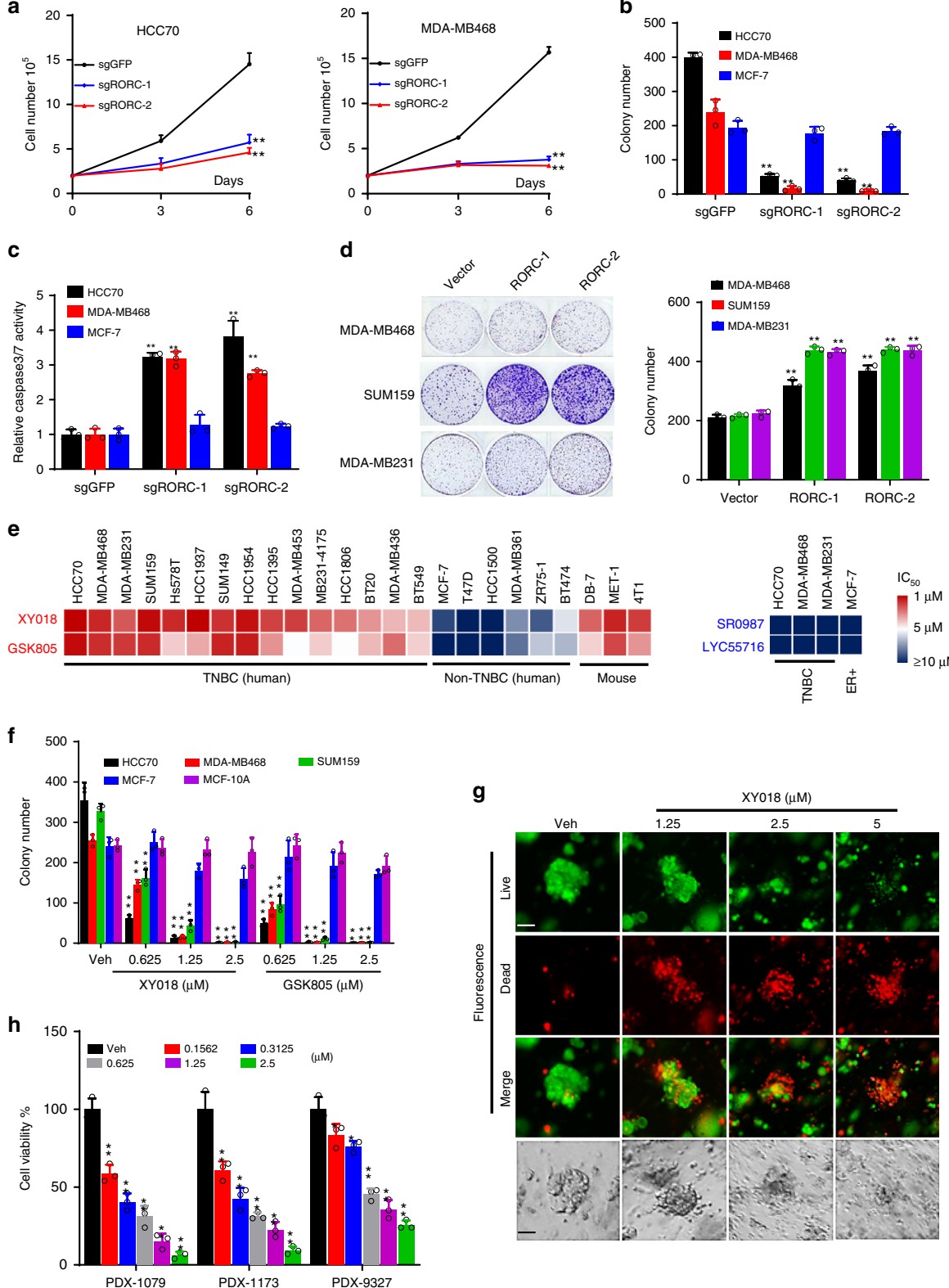

the survival of TNBC but not that of ER+ cells were observed in cell number and colony-formation assays (Fig. 2f, Supplementary Fig. 2i, j). In line with the apoptotic effects by RORC gene silencing, XY018 and GSK805 strongly elevated cleaved PARP-1 protein level and the activity of caspase3/7 enzymes and markedly reduced key proliferation and survival protein expression in the TNBC cells (Supplementary Fig. 2k,

l). Given that 3D organoids may closely mimic clinical tumors in response to therapeutics, we treated organoids derived from three different TNBC PDX tumors with the antagonist. We found that all three organoids were highly sensitive to XY018 treatment (Fig. 2g, h). Together, these results strongly support the notion that RORγ is a major driver of TNBC cell survival.

**Fig. 2** RORγ is a major driver of TNBC cell survival. **a** TNBC cells were infected with lentiviruses expressing control sgRNA against GFP or two different sgRNAs against RORC and Cas9. Three and six days later, viable cell numbers were counted. **b** TNBC cells were infected as in (**a**). Fourteen days later, colonies were counted. **c** Caspase 3/7 activities were measured by using a luminescent caspase-Glo 3/7 assay kit with TNBC cells harvested 3 days after the infections as in (**a**). **d** TNBC cells were infected by RORγ overexpression or control lentivirus. Fourteen days later, representative images of colony formation were taken (left) and colonies were counted (right). **e** Heat map presentation of IC$_{50}$ for RORγ antagonists GSK805 and XY018, or agonists SR0987 and LYC55716 in indicated cell lines treated for 4 days. Cell viability was measured with CellTiter Glo on GLOMAX microplate luminometer. **f** Indicated cells were treated with vehicle (DMSO) or different concentrations of XY018 and GSK805 for 14 days, after which colonies was counted. **g**, **h** PDX-derived organoids were treated with DMSO or indicated concentrations of XY018. **g** Six days later, representative images were taken under a fluorescence microscope (top three rows) or standard light microscope (bottom row). Scale bar represents 20 μM. **h** Four days later, cell viability in organoids was measured with CellTiter-Glo. Data are shown as mean ± s.d. $n = 3$. Student's $t$ test. $**p < 0.01$

**RORγ is a master activator of cholesterol biosynthesis.** To identify the core transcriptional program controlled by RORγ in TNBC cells, we first performed RNA-seq analysis of HCC70 and MDA-MB468 cells treated with antagonists XY018 or GSK805. Gene ontology (GO) analysis of the 159 transcripts commonly downregulated by the antagonists in the two TNBC cells revealed that genes involved in cholesterol-biosynthesis pathway were among the most highly enriched (Fig. 3a, b). Further examination by gene-set enrichment analysis (GSEA) also indicated clearly that hallmarks of cholesterol-biosynthesis pathway were strongly altered by XY018 or GSK805 (Fig. 3c, Supplementary Fig. 3a). Cholesterol-biosynthesis program involves at least 21 genes encoding enzymes to convert acetyl-CoA into cholesterol (Supplementary Fig. 3b). Gene-encoding rate-limiting enzymes HMGCR and SQLE are commonly deregulated in cancer. Our pathway-focused analysis demonstrated that the vast majority of the cholesterol-biosynthesis genes were significantly downregulated by the RORγ antagonists in a dose-dependent manner in both TNBC cell lines and tumor organoids (Fig. 3d left two panels). In contrast, ectopic RORγ strongly increased the expression of the majority of the pathway genes including *HMGCS1*, *MVK*, and *SQLE* (Fig. 3d, right panel). Consistently, treatment with the RORγ antagonists resulted in a strong downregulation of the proteins including HMGCS1, HMGCR, MVK, and SQLE (Fig. 3e). Moreover, siRNA silencing of RORγ also led to a similar downregulation of the cholesterol-biosynthesis genes and proteins (Supplementary Fig. 3c). In line with the lack of strong growth or survival-promoting role by RORγ in ER+ cells, treatment of MCF-7 cells with the antagonist did not cause any significant decreases (Supplementary Fig. 3d). Importantly, consistent with the effects on gene expression, inhibition of RORγ by its selective antagonists significantly decreased cellular cholesterol content in the TNBC cells. Likewise, RORγ silencing also reduced the cholesterol content (Fig. 3f, Supplementary Fig. 3e). Together, these results support a notion that RORγ is a previously unrecognized transcriptional activator of cholesterol-biosynthesis program in TNBC.

**RORγ plays a dominant role over SREBP2.** As expected, transient, ectopic expression of the nuclear form SREBP2 effectively activated the cholesterol-biosynthesis genes (Fig. 3g). However, treating the cells with the RORγ antagonist after the ectopic expression not only eliminated the nSREBP2-induced activation but also resulted in a net repression similar to that observed in cells only treated with the antagonist, suggesting that RORγ inhibition is sufficient to completely negate the activation activity of SREBP2. To examine whether the reverse is true, we silenced SREBP2 expression with siRNA in cells that ectopically overexpressed RORγ. Interestingly, knockdown of SREBP2 had essentially no effect on the activation of the cholesterol-biosynthesis program by the RORγ (Fig. 3h), suggesting a SREBP2-independent function of an overexpressed RORγ.

Notably, the antagonists did not affect the level of SREBP2 mRNA or the transcriptionally active, nuclear form SREBP2 protein (Supplementary Fig. 3f, g), suggesting that RORγ may regulate SREBP2 function at a chromatin level.

**RORγ mediates SREBP2 chromatin recruitment and activation.** To define the mechanism underlying the dominant function of RORγ over that of SREBP2, we performed ChIP-seq analysis of RORγ and SREBP2 genome-wide binding in HCC70 cells. The analysis identified a total of 31,543 high-confidence binding sites for RORγ and 48,144 binding sites for SREBP2. Interestingly, the binding sites of the two proteins are largely overlapping with 86.8% of the RORγ-binding sites shared by SREBP2 and 56.9% of SREBP2-binding sites co-bound by RORγ (Fig. 4a). GO analysis of genes that displayed ChIP-seq peaks for both RORγ and SREBP2 in HCC70 cells revealed that steroid/cortisol synthesis was among the significantly enriched programs (Supplementary Fig. 4a, b). Transcription factor motif analysis of SREBP2 peak regions identified ROREs as one of the top-ranking motifs (Fig. 4c). Treatment of TNBC cells with the RORγ antagonist markedly reduced genome-wide SREBP2 association with its target loci, while little effect was observed in RORγ genome-wide binding (Fig. 4d). In line with the crucial role of RORγ in controlling cholesterol-biosynthesis program, RORγ antagonists drastically diminished SREBP2 binding to most of the program gene loci (Fig. 4e top, Supplementary Fig. 4b left). Indeed, strong reduction of SREBP2 binding was observed at promoters of the major targets such as *HMGCS1, MVK, SQLE*, and *DHCR24* in ChIP-seq and ChIP-qPCR analyses (Fig. 4f top, Supplementary Fig. 4c, d top). Concomitant with the loss of SREBP2 occupancy, the transcriptional activation-linked histone mark H3K27ac was also significantly reduced at the chromatin regions (Fig. 4d–f bottom, Supplementary Fig. 4b right, d bottom, e). The occupancy of SREBP2 coactivator histone acetylase p300 was also significantly decreased by the antagonist (Supplementary Fig. 4e). In line with the reduction of mRNA levels of cholesterol-biosynthesis genes, promoter occupancies of RNA Polymerase II (Pol-II), particularly its transcription initiation-associated CTD-ser 5 phosphorylated form (S5P Pol-II), were also reduced in the antagonist-treated TNBC cells.

Next, we examined whether and, if yes, how at the cholesterol-biosynthesis gene promoter RORγ functions with SREBP2. Consistent with their genome-wide co-distribution, overlapping of RORγ- and SREBP2-binding sites was also observed at the majority of cholesterol-biosynthesis genes (Supplementary Fig. 4f). ChIP-seq and ChIP-qPCR analyses demonstrated that indeed RORγ was recruited to the putative RORE sites at cholesterol-biosynthesis gene promoter regions containing SREBP2-binding sites (Fig. 4f middle, Supplementary Fig. 4d middle, g). We then performed reporter-gene assays with promoters of *MVK* and *HMGCS1* and found that they were highly responsive to RORγ-mediated transactivation. Mutations

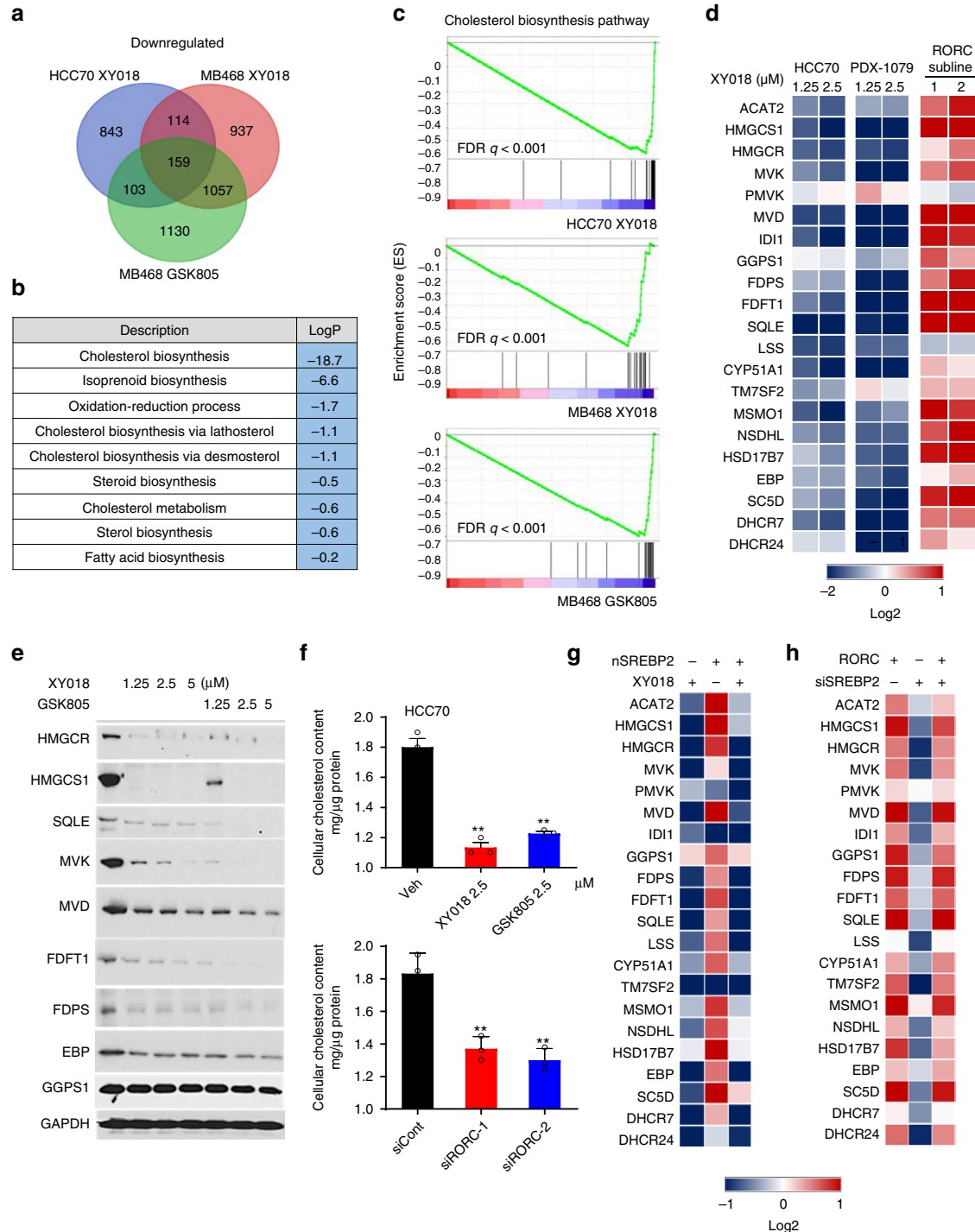

of the putative ROREs or treatment with RORγ antagonists effectively diminished the RORγ-dependent activation (Fig. 4g, Supplementary Fig. 4h). Together, the data indicate that indeed RORγ activated SREBP2 target promoters via the putative ROREs. We then asked whether RORγ can act as a mediator of SREBP2 transcriptional activation. Co-transfection experiments show that simultaneous expression of RORγ and nSREBP resulted in a synergistic transactivation of the *MVK* gene promoter, suggesting that RORγ can act as a coactivator of SREBP2 (Fig. 4h). Finally, we performed co-immunoprecipitation by using MDA-MB468 cells expressing V5-RORγ and Flag-nSREBP2 and found that nSREBP2 protein was readily detected in the immune precipitates of RORγ. Interestingly, the association could be largely diminished by treating cells with XY018 (Fig. 4i). Taken together, the results suggest that in TNBC cells, RORγ plays a dominant role in activating cholesterol-biosynthesis genes via its own binding to the ROREs at the targets, which in turn facilitates the recruitment of nSREBP2 to further stimulate the cholesterol-biosynthesis gene transcription.

**RORγ inhibitors synergize with statins in killing TNBC cells.** The prominent role of RORγ in controlling cholesterol-biosynthesis program prompted us to first examine whether its

**Fig. 3** RORγ is a master activator of cholesterol biosynthesis in TNBC. **a** Venn diagram of the number of genes with expression significantly (1.5-fold) downregulated, which is detected by RNA-seq of HCC70 and MDA-MB468 cells treated for 24 h with 2.5 μM XY018 or GSK805. **b** Gene ontology analysis of the 159 genes with expression downregulated in both HCC70 and MB468 cells by XY018 and GSK805 treatment as shown in (**a**). Hypergeometric test and Benjamini–Hochberg p-value correction. **c** GSEA plots depicting the enrichment of genes downregulated (1.5-fold) in cholesterol-biosynthesis pathway in HCC70 cells treated with XY018 (top), or in MDA-MB468 cells treated with XY018 (middle) or GSK805 (bottom). FDR false-discovery rate. **d** Heat maps of mRNA expression changes of 21 cholesterol-biosynthesis genes in TNBC cells. Left, mRNA expression of HCC70 treated with XY018 for 24 h was analyzed by RNA-seq. Middle, mRNA expression of organoids derived from PDX-1079 treated with XY018 for 24 h was analyzed by qRT-PCR. Right, mRNA expression of MDA-MB468 cells treated with vector control or overexpressed RORC was analyzed by qRT-PCR. n = 3. The experiments were repeated three times. **e** Immunoblotting of proteins involved in cholesterol-biosynthesis pathway in MDA-MB468 cells treated with XY018 or GSK805 for 3 days. The experiments were repeated three times. **f** Total cellular cholesterol contents in HCC70 cells treated with indicated RORγ antagonists (top) or transfected with siRORC (bottom) for 3 days were analyzed after organic extraction and normalization to protein concentrations. n = 3, data are shown as mean ± s.d. Student's t test. **p < 0.01. **g** Heat map display of fold changes (in log2) of cholesterol-biosynthesis pathway gene mRNA analyzed by qRT-PCR in HCC70 cells transfected with nSREBP2 or control vector for 24 h and then treated with XY018 (2.5 μM) or vehicle (DMSO) for another 24 h. n = 3. The experiments were repeated three times. **h** Heat map display of fold changes (in log2) of cholesterol-biosynthesis pathway gene mRNA analyzed by qRT-PCR in RORC-overexpressed MDA-MB468 cells transfected with siSREBP2 or control siRNA for 24 h, n = 3. The experiments were repeated three times. Source data are provided in a Source Data file

antagonistic effect on TNBC cell survival is attributable to cholesterol depletion. Indeed, exogenous cholesterol supply after the antagonist treatment was effective in preventing cell number reduction by the two antagonists (Fig. 5a, Supplementary Fig. 5a). Different from the cholesterol-lowering effect by 2.5 μM XY018 (Fig. 3f, Supplementary Fig. 3e), a lower concentration of XY018 (1.25 μM) had only a marginal effect on the cholesterol content in the TNBC cells (Fig. 5b, Supplementary Fig. 5b). Likewise, 1.25 μM ATV also had no significant effect. However, a combination of 1.25 μM XY018 and ATV caused a pronounced reduction of cholesterol content. To examine whether RORγ inhibition can also synergize with statins in killing TNBC cells, we treated TNBC cells with a combination of different concentrations of RORγ inhibitors and statins. Simultaneous treatment of TNBC cells with increasing concentrations of ATV and XY018 elicited a highly synergistic inhibition of cell growth (Fig. 5c, Supplementary Fig. 5c). As expected, a combined treatment of TNBC cells (HCC70 or MB468) with relatively low concentrations of RORγ antagonists and statins dramatically reduced their colony formations (Fig. 5d, Supplementary Fig. 5d). In contrast, similar combinations did not significantly affect the colony formations or cell growth in the MCF-7 cells. Treatment of multiple TNBC organoid cultures also demonstrated a strong synergistic effect on cell viability (Fig. 5e, Supplementary Fig. 5e). Different statins may have different efficacy in inhibition of cancer cell growth and survival[29]. When combined with low concentrations of GSK805 or XY018 (at about 1/5 of their IC$_{50}$), the IC$_{50}$ values of the widely used statins (i.e., ATV, simvastatin/SIM, lovastatin, fluvastatin, and rosuvastatin) all significantly dropped approximately three-to tenfold in all three TNBC cells examined (Supplementary Fig. 5f), indicating that the growth inhibition synergy is not specific to a particular statin drug. Moreover, treating cells with a low concentration of RORC-specific siRNAs, was effective in sensitizing TNBC cells to statin (Supplementary Fig. 5g), suggesting that RORγ inhibition-induced synergy with statin is not limited to a specific RORγ antagonist.

**RORγ inhibitors negate statin-induced feedback**. It has been postulated that strong induction of SREBP2-dependent feedback gene activation may underlie the lack of sensitivity to statins by cancer cells/tumors[1]. Indeed, our RNA-seq analysis revealed that cholesterol and sterol biosynthesis pathways are among the most enriched programs upregulated by statin treatment of TNBC cells (Supplementary Fig. 5h). We thus reasoned that mitigating statin-induced feedback might constitute a major mechanism of the synergy by the combination of statin with RORγ antagonists. We

analyzed gene expressions altered by their singular or combined treatments using GO and GSEA and found that the cholesterol/isoprenoid-biosynthesis program is the most significantly upregulated by statin alone, but downregulated by the RORγ antagonists when either alone or in combination with statin (Fig. 5f–h). Our pathway-focused RNA-seq analysis demonstrated that indeed co-treatment of TNBC cells with RORγ antagonists and statin completely abolished the induction by statin alone of cholesterol-biosynthesis genes (Fig. 5i). Notably, the co-treatment resulted in a net decrease of the core cholesterol-biosynthesis enzyme proteins (Fig. 5j). Such superior inhibition was also observed in the TNBC organoids (Supplementary Fig. 5i). Conversely, even though statin induced a similar adaptive activation of cholesterol-biosynthesis program in ER+ cells, the RORγ antagonist alone only dampened the activation slightly. Its combination with statin also did not cause any significant net inhibition (Supplementary Fig. 5j). Taken together, the results support a notion that inhibition of RORγ can completely negate statin-induced negative feedback and result in strong synergy with statin in killing TNBC cells.

**RORγ inhibitors cause tumor regression and block metastasis**. To examine the therapeutic value of RORγ antagonists, we first treated mice bearing MB468-derived orthotopic xenograft tumors, i.p., with two different doses of XY018. We found that at a relatively low dose of 2.5 mg/kg, the antagonist was effective in inhibition of the tumor growth, while a higher dose (5 mg/kg) displayed a complete tumor growth blockade for over 7 weeks (Supplementary Fig. 6a). Similar potent tumor-inhibition activities of antagonists GSK805 and XY018 at 5 mg/kg were observed in additional TNBC HCC70 and SUM159 cell-derived xenografts (Supplementary Fig. 6b, c). To provide data more relevant in a clinical setting, we measured the oral dosing (p.o.) efficacy of XY018 in a PDX model of TNBC and found that oral administration of XY018 (50 mg/kg) resulted in a tumor regression (Fig. 6a, Supplementary Fig. 6d). In line with a lack of significant effects on ER+ cell growth, XY018 treatment via i.p. (10 mg/kg) did not suppress the growth of MCF-7-derived orthotopic xenografts (Supplementary Fig. 6e).

Next, we examined whether RORγ antagonists have effects on tumor growth and metastasis in an immune-intact host environment with 4T1 syngeneic tumors. Similar to the effect in the xenograft models, the RORγ antagonist strongly inhibited 4T1 tumor growth, and more importantly, dramatically extended the host animal survival (Fig. 6b, Supplementary Fig. 6f). It is well known that the 4T1 primary tumor cells at the mammary gland

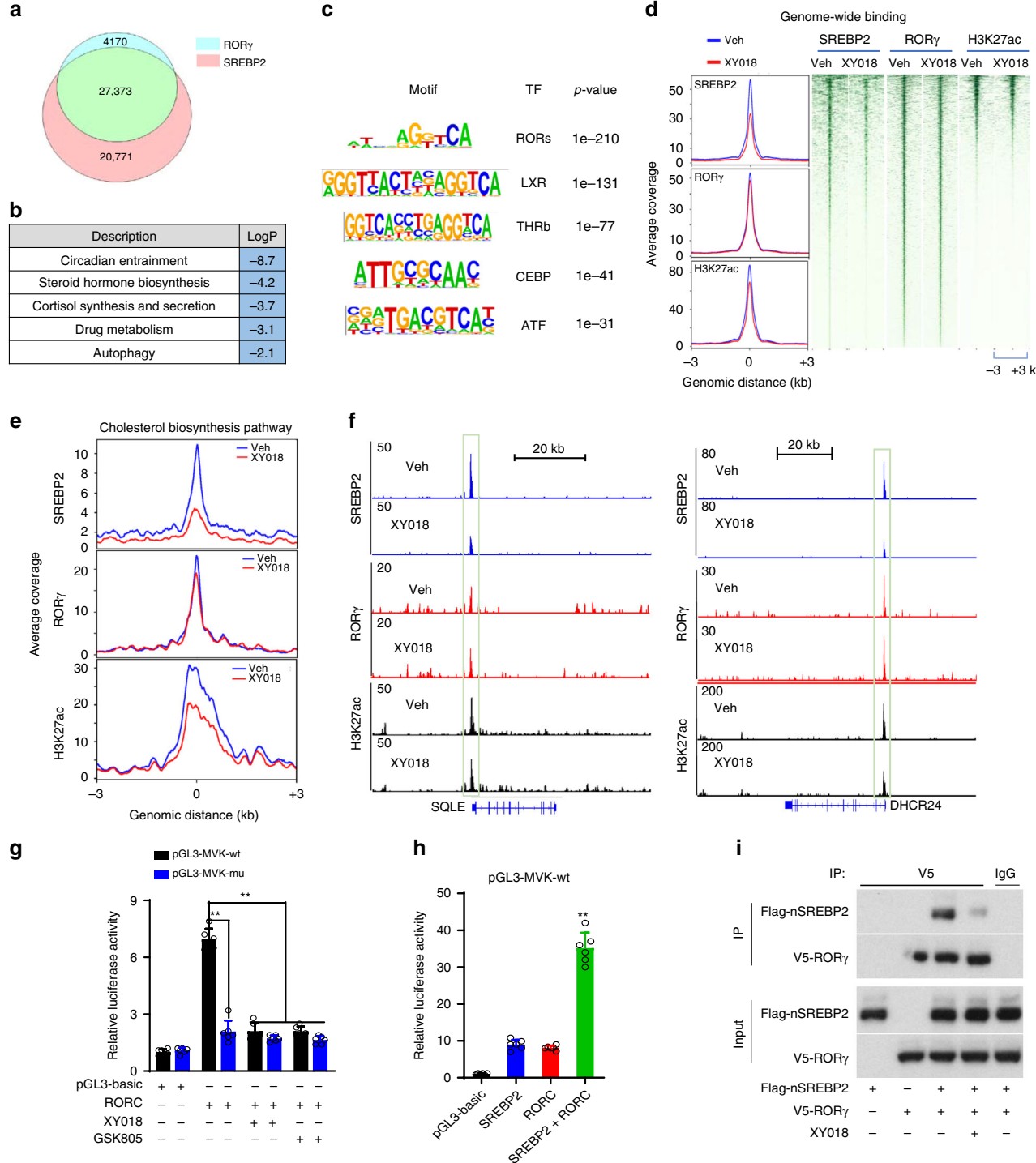

metastasize with high efficiency to the lung. To examine the potential effect of the antagonist on tumor metastasis, we surgically removed the primary tumors at mammary glands and then treated the mice with XY018 daily for 4 weeks. Histological analysis revealed that tumor nodules formed in the lung were significantly reduced (Fig. 6c). To further establish the anti-metastasis activity of targeting RORγ, we used a lung-metastatic model (LM2) of MDA-MB231. After the animals received LM2 cells via the tail vein, we treated them with XY018 for 4 weeks before measuring tumor cell bioluminescence and observed a strong reduction of LM2 lung colonization in the treated mice (Supplementary Fig. 6g).

**RORγ inhibitors in combination with statins regress tumors.** We next evaluated the therapeutic potential of combining RORγ inhibitors with statins in multiple TNBC models. First, in two PDX models (1079 and 1173), oral administration of antagonist XY018 alone (20 mg/kg) significantly inhibited the tumor growth, whereas ATV alone (15 mg/kg, p.o.) did not display any significant effect. Remarkably, their combined treatment caused a tumor regression (Fig. 6d, e, Supplementary Fig. 6h, i). Moreover, in PDX-1079-bearing animals, combined treatment of the RORγ antagonist with a different statin drug (SIM) elicited essentially the same strong synergy (Fig. 6f, Supplementary Fig. 6j). Finally, in the MDA-MB468 cell-derived xenografts, the combined

**Fig. 4** RORγ plays a dominant role over SREBP2 to reprogram cholesterol biosynthesis. **a** Venn diagram of the number of high-confidence (FDR 1% and IDR 1%) binding sites shared by RORγ ChIP-seq and SREBP2 ChIP-seq in HCC70 cells. FDR false-discovery rate, IDR irreproducible discovery rate. **b** Gene ontology analysis of genes that displayed ChIP-seq peaks for both RORγ and SREBP2 in HCC70 cells. Gene list of each program is shown in Supplementary Table 1. Hypergeometric and binomial test $p$ value. **c** Transcription factor (TF) motifs that are enriched in chromatin regions with SREBP2 peaks diminished by 2.5 μM XY018 treatment of HCC70 for 24 h. **d** ChIP-seq profiles (left) and heat maps of ChIP-Seq signal intensity (right) of SREBP2, RORγ, or H3K27ac within ±3-kb windows around the center of peak regions in HCC70 cells treated with 2.5 μM XY018 or vehicle for 24 h. **e** ChIP-Seq profiles of SREBP2 (top), RORγ (middle), or H3K27ac (bottom) binding within ±3-kb windows around the center of peak regions on genes involved in cholesterol-biosynthesis pathway in HCC70 cells treated as in (**d**). **f** ChIP-seq signal visualization of SREBP2 (top), RORγ (middle), or H3K27ac (bottom) at representative cholesterol-biosynthesis genes *SQLE* and *DHCR24* in HCC70 cells treated as in (**d**). **g** *MVK* (wild type or RORE mutated) promoter luciferase reporter activity changes by RORγ overexpression or treatment with 2.5 μM RORγ antagonists XY018 and GSK805 in MDA-MB468 cells for 24 h. Data are shown as mean ± s.d. $n = 6$. Student's $t$ test. **$p < 0.01$. **h** *MVK* promoter luciferase reporter activity changes in HEK293T cells co-transfected with RORC, nSREBP2, or both for 24 h. $n = 6$. Data are shown as mean ± s.d. Student's $t$ test. **$p < 0.01$. **i** Co-IP analysis with V5-RORC expressing MDA-MB468 cells transfected with Flag-nSREBP2 expression vector. Twenty-four hours after transfection, cells were treated with 2.5 μM XY018 or vehicle for another 24 h. The nuclear extracts were used for co-IP with indicated antibodies. The experiments were repeated three times. Source data are provided in a Source Data file

treatment also displayed a synergistic tumor growth inhibition (Fig. 6g, Supplementary Fig. 6k). Importantly, neither the single nor the combined treatments affected the animal body weight over the course of the treatment, suggesting that the RORγ inhibitor or the combined treatment strategy is relatively safe. Together, the results from our animal experiments strongly suggest that RORγ inhibitor alone or its combination with statin can be a new strategy for effective treatment of TNBC.

**RORγ antagonists inhibit TNBC tumor cholesterol biosynthesis.** Having demonstrated the critical role of RORγ in reprogramming of TNBC de novo cholesterol biosynthesis in vitro, we sought to examine whether RORγ antagonists inhibit tumor growth through blocking aforementioned reprogramming. We first measured tumor cholesterol content at different time points after the treatments. Consistent with the in vitro effects, we observed a marked decrease of cholesterol content in tumors from mice treated with XY018 or a combination of XY018 and ATV, for either 2 or 7 days, but not in tumors treated with ATV alone (Fig. 7a). Notably, co-treatment with XY018 and ATV caused a significantly greater reduction of tumor cholesterol content when compared with that of XY018 alone. To determine whether the cholesterol content reduction effects can be attributed to decreased cholesterol biosynthesis in vivo, we measured the rate of cholesterol biosynthesis in tumors and in the livers of host animals by using gas chromatography–mass spectrometry to quantify the abundance of deuterated hydrogen atoms incorporated in the newly synthesized cholesterol molecules. In agreement with the reduced cholesterol content in tumors, single-agent XY018 significantly decreased cholesterol biosynthesis rate in the tumors. XY018 and ATV-combined treatment caused significant reduction more than that of the single agent (Fig. 7b). Consistent with the rate decrease, tumor mRNA and protein expression of HMGCS1, MVK, MVD, and SQLE was also significantly down-regulated by XY018 or its combination with statin (Fig. 7c, Supplementary Fig. 7a). Interestingly, tumor LDLR and ABCA1 expression was not significantly changed by the RORγ antagonist. Together, the results support an interpretation that in TNBC tumor, inhibition of de novo cholesterol biosynthesis is a predominant action mechanism of the RORγ inhibitors.

**RORγ antagonists preserve host cholesterol homeostasis.** To maintain whole-body cholesterol homeostasis, organs such as the liver play a major role. Thus, we also measured cholesterol contents of the mouse liver and found that, contrary to the effects in tumors, the RORγ inhibitor treatments, either alone or in combination with statin, did not significantly alter liver cholesterol

content (Fig. 7d). However, as reported in previous studies[30], statin markedly increased the cholesterol-biosynthesis rate in the liver (Fig. 7e). Consistent with the rate increase, statin strongly induced liver expression of most of the cholesterol-biosynthesis genes (Fig. 7f). In contrast to its effects in the tumors, the RORγ antagonist alone or in combination with ATV did not display any significant inhibition of liver cholesterol-biosynthesis gene expression (Fig. 7f). Also as reported[30,31], statin induced a reduction of circulating cholesterol. However, such reduction was not affected by the antagonist (Fig. 7g). Together, these data indicate that treatment of animals with the RORγ antagonist does not have any significant effect on normal liver function of maintaining cholesterol homeostasis. The results also point to a clear distinction in the function of RORγ between TNBC tumors and the host liver tissue.

To evaluate the overall safety profile of the antagonist and its combination with statin, we measured the major blood biochemical and cell parameters in MDA-MB468-bearing SCID mice after 63 days of the treatments. The results suggest that the treatment, either alone or with statin, did not cause any significant impairment of complete cell count or function of liver and kidney (Supplementary Fig. 7b, c). Furthermore, single or combined therapy did not show any significant impacts on liver histology and organ weight during the course of the treatment (Supplementary Fig. 7d, e). Together, the data suggest that treatment with RORγ antagonist alone or in combination can be efficacious in blocking tumor growth and metastasis without causing any overt toxicity.

## Discussion

Cancer metabolic reprogramming is often context specific[5,32–35]. Here, we show that TNBC exhibits a hyper-activated gene expression profile of the cholesterol-biosynthesis pathway, which is even higher than that of ER+ breast cancer. Our search for possible factors responsible for the distinction led to the unearthing of RORγ as a master regulator of this metabolic pathway in TNBC. We demonstrate that RORγ controls virtually all of the cholesterol-biosynthesis pathway genes that are targets of SREBP2. Unexpectedly, we found that this function of RORγ is dominant over that of SREBP2 and that it operates in TNBC but not in ER+ breast cancer. Together, with the vital role shown by RORγ in TNBC cell growth and survival and the pronounced effects displayed by the RORγ antagonists on tumor growth and metastasis, this study revealed an NR as a previously unsuspected novel driver of tumor subtype-specific metabolic reprogramming and an attractive target for TNBC.

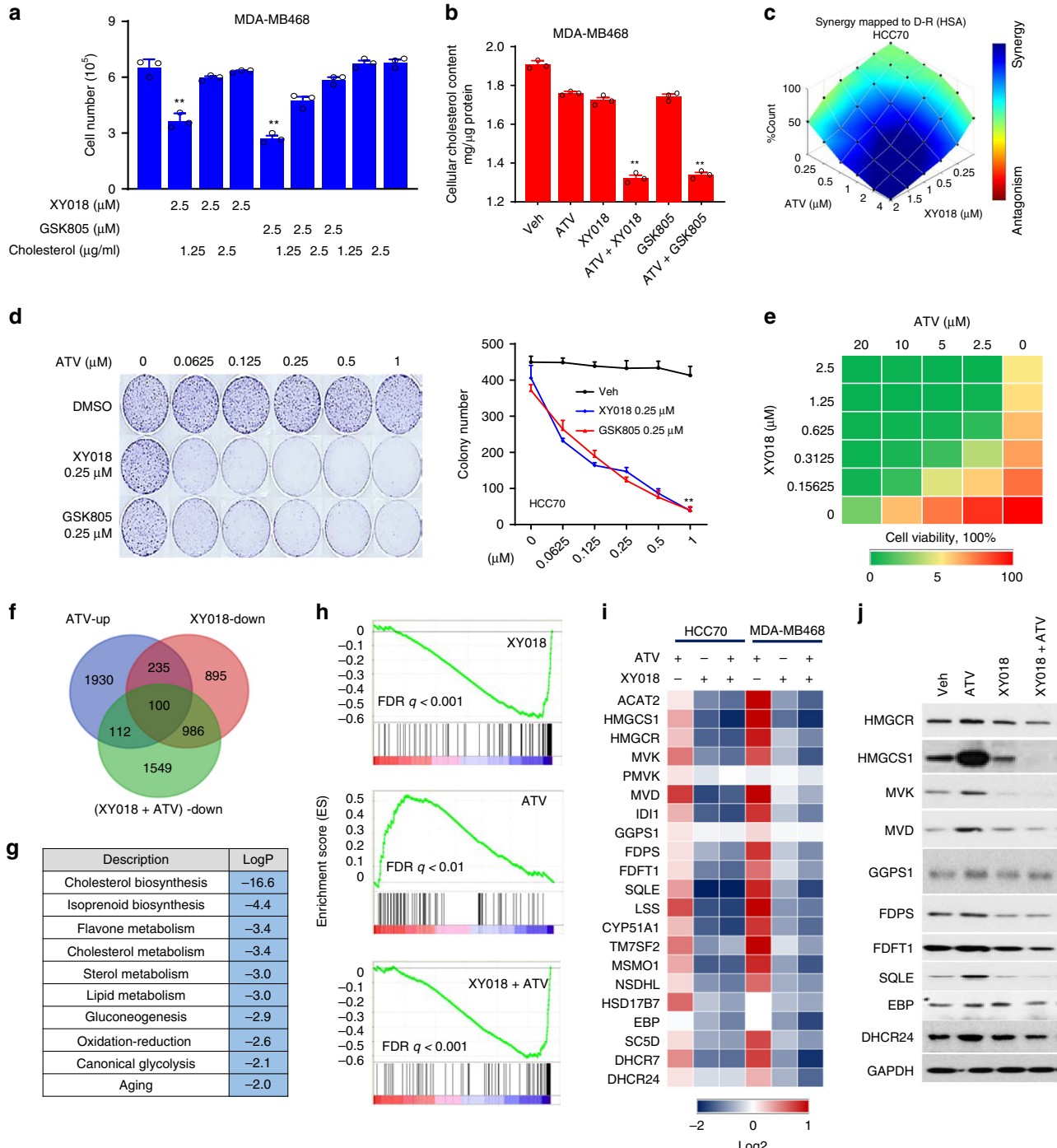

**Fig. 5** RORγ inhibitors strongly synergize with statins in killing TNBC cells. **a** Exogenous cholesterol supply for 24 h rescued MDA-MB468 cell death caused by treatment with RORγ antagonists for 24 h. $n = 3$. **b** Total cellular cholesterol contents in MDA-MB468 cells treated with indicated RORγ antagonists (1.25 μM) or atorvastatin (ATV, 1.25 μM) for 2 days were analyzed after organic extraction. $n = 3$. **c** HCC70 cells were treated with XY018 and ATV as indicated for 2 days. Cell numbers were counted. Blue indicates synergy while red indicates antagonism between drugs. $n = 3$, the experiments were repeated three times. **d** HCC70 cells were treated with indicated concentrations of XY018 or GSK805 alone, or in combination with ATV for 2 weeks. Representative images of colony formation were taken (left), and colonies were counted (right). $n = 3$. **e** Organoids derived from PDX-1079 were treated with XY018 and ATV as indicated for 3 days. The cell viability was measured with CellTiter-Glo. $n = 3$. The experiments were repeated three times. **f** Venn diagram of the number of genes significantly downregulated by XY018 (1.25 μM), or upregulated by ATV (1.25 μM) or downregulated by XY018 + ATV in HCC70 cells treated for 24 h, which were detected by RNA-seq. **g** Gene ontology analysis of the 100 genes overlapped in expression alterations as shown in (**f**) in response to indicated compound treatment. Hypergeometric test and Benjamini–Hochberg *p*-value correction. **h** GSEA plots depicting the enrichment of genes downregulated (1.5-fold) in cholesterol-biosynthesis pathway in HCC70 cells treated with XY018 (up) or ATV (middle) or both XY018 and ATV (bottom). FDR false-discovery rate. **i** Heat maps of mRNA expression changes of cholesterol-biosynthesis genes in cells treated as indicated for 24 h and analyzed by RNA-seq. **j** Immunoblotting of proteins involved in cholesterol-biosynthesis pathway in MDA-MB468 cells treated with vehicle, 2.5 μM XY018, 1.25 μM ATV, or both XY018 and ATV for 3 days. The experiments were repeated three times. Data are shown as mean ± s.d. Student's *t* test. **$p < 0.01$. Source data are provided in a Source Data file

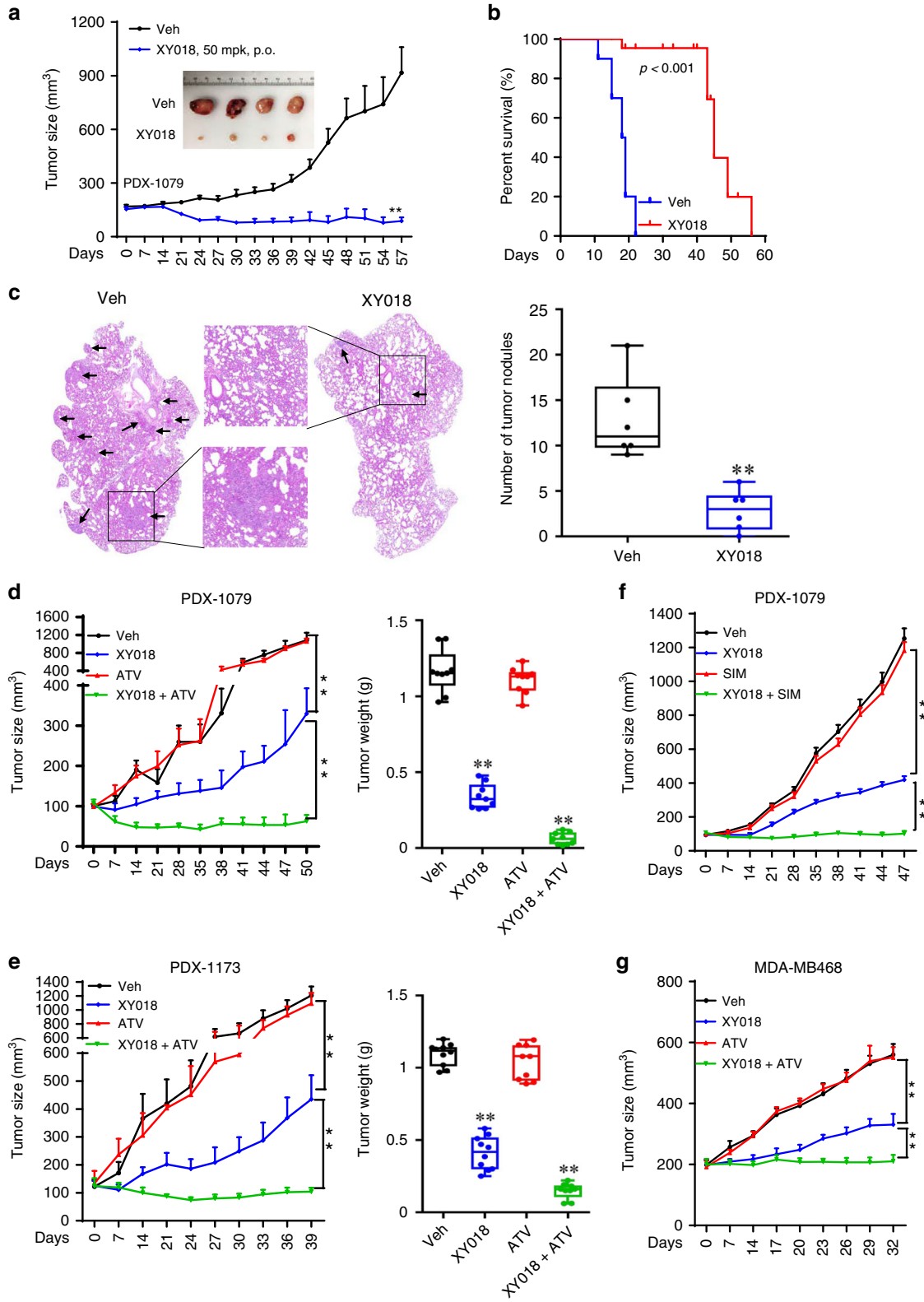

Despite their heterogeneity at molecular and pathological levels, most TNBC tumors feature a high proliferative index. Thus, it is likely that a hyper-activated cholesterol-biosynthesis program is needed to feed the high demand of membrane lipids and other metabolites by TNBC tumors for their rapid growth. The basis for TNBCs to co-opt an activator like RORγ for the hyper-activation, instead of revamping the established master

SREBP2, is unclear at this point. One advantage might be that adoption of RORγ can better meet the unique need of TNBC and at the same time be independent of SREBP-mediated feedback that is hard-wired in highly differentiated tissues such as the liver. In support of this notion, first, we observed that RORγ plays a dominant function over that of SREBP2, as overexpressed RORγ can activate cholesterol-biosynthesis program independent of

**Fig. 6** RORγ inhibitors alone or with statins cause tumor regression and block metastasis. **a** Mice bearing PDX-1079 were treated, p.o., 5 times per week, with vehicle or 50 mg/kg XY018 for 57 days. Tumor volume and representative images are shown. $n = 7$ mice per group. Student's $t$ test. **$p < 0.01$. mpk, mg/kg. **b** K–M survival plot of 4T1 tumor-bearing mice treated with vehicle or XY018 (20 mg/kg, i.p.) for 56 days, $n = 10$. **c** Lung tumor nodules metastasized from primary sites were analyzed for 4T1 tumor-carrying mice treated with XY018 (20 mg/kg, i.p.) or vehicle daily for 4 weeks. Representative lung section images were taken with arrows indicating tumor nodules. $n = 6$. Student's $t$ test. **$p < 0.01$. **d**, **e** PDX-1079 and PDX-1173 were treated, 5 times per week, p.o., with vehicle, 20 mg/kg XY018 alone, 15 mg/kg ATV alone, or both XY018 + ATV for indicated days. Volumes (left) and weights (right) are shown. $n = 7$. Student's $t$ test. **$p < 0.01$. **f** PDX-1079 was treated, 5 times per week, p.o., with vehicle, 20 mg/kg XY018 alone, 15 mg/kg simvastatin alone (SIM), or both XY018 + SIM for 47 days. Volumes are shown. $n = 7$. Student's $t$ test. **$p < 0.01$. **g** Mice bearing MDA-MB468 were treated, 5 times per week, p.o., with vehicle, 20 mg/kg XY018 alone, 15 mg/kg ATV alone, or both XY018 + ATV for 32 days. Volumes are shown. $n = 7$. Student's $t$ test. **$p < 0.01$

SREBP2, whereas the function of SREBP2 is mediated by RORγ. Second, we found that RORγ does not mediate statin-induced feedback for cholesterol- biosynthesis stimulation. Moreover, the choice of RORγ as a new driver of cholesterol biosynthesis might also be attributable to the possible unique mode of regulation of its function. Recent studies show that RORγ can bind to and be further activated by the intermediates of cholesterol biosynthesis[23]. Therefore, a logical hypothesis will be that by co-opting RORγ, TNBC cells can effectively enforce their cholesterol-biosynthesis program with a feed-forward loop. Different from TNBC, in estrogen-sensitive ER+ tumors, the cholesterol-biosynthesis program also produces steroid hormones such as 17-β estradiol (E2) to activate ERα for its mitogenic function. In endocrine therapy-resistant ER+ tumors, cholesterol metabolites such as 27-hydroxycholesterol (27-HC) can replace E2 to activate ERα[36–38]. Therefore, it is possible that in ER+ tumors, ERα can be a key determinant in cholesterol-biosynthesis reprogramming.

Echoing the TNBC addiction to the function of RORγ, RORγ-selective antagonists were highly potent in causing tumor regression and blocking metastasis of multiple TNBC models, but not of ER+ tumors. Although this study focused largely on tumor cell-autonomous role of RORγ, our results do not exclude the possible function of RORγ in the tumor microenvironment. RORγt, the T-cell-specific isoform, is instrumental in eliciting a proinflammatory program in multiple lymphocyte lineages including T helper type 17 (Th17) and γδT cells. In multiple preclinical models, including 4T1 tumors, tumor-infiltrating IL-17-producing γδT and Th17 cells show potent protumoral functions through their recruitment or interactions with other immune cells such as neutrophils, MDSCs, M2 macrophages, and Treg cells, which can lead to immune suppression, enhanced angiogenesis, and metastasis[39–43]. Importantly, high levels of IL-17-producing cell infiltrations are associated with the triple-negative status and shorter disease-free survival[39,41,44]. In line with this, we observed that the RORγ antagonists were also effective in suppressing metastasis and enhancing survival in the 4T1 immune-intact model. Moreover, a recent study has shown that inhibition of cholesterol-biosynthesis pathway induces T-cell responses and enhances antigen-specific antitumor immunity[45].

The attractiveness of RORγ inhibition as a targeted therapy for TNBC is also reflected by a lack of significant impact on host lipid homeostasis. Statin use at doses for treatment of hypercholesterolemia is generally safe. However, its efficacy in an adjuvant setting as an anticancer drug appears very limited in clinical trials likely because of the SREBP2-mediated tumor feedback response[18]. The strong anti-TNBC synergism displayed by a combination of RORγ antagonist and statin is a clear manifestation of a potent blockade of the feedback by RORγ inhibition. In conclusion, our results collectively indicate that the unique function of RORγ offers a new opportunity for effective treatment of a cancer subtype with addiction to a hyper-activated cholesterol-biosynthesis program.

## Methods

**Cell lines**. HCC70, HCC1500, HCC1937, HCC1806, HCC1935, HCC1954, and ZR75-1 cells were cultured in RPMI1640 supplemented with 10% FBS. MDA-MB231, MDA-MB468, MDA-MB436, MDA-MB453, MDA-MD231-derived LM2 (4175), MCF-7, MDA-MB361, BT474, and BT20 were cultured in DMEM supplemented with 10% FBS. T47D and BT549 were cultured in RPMI1640 supplemented with 10% FBS and insulin (10 μg/ml). Hs578T was cultured in DMEM supplemented with 10% FBS and insulin (10 μg/ml). SUM149 and SUM159 were cultured in DMEM/F12 medium supplemented with 5% FBS and insulin (10 μg/ml) and hydrocortisone (0.5 mg/ml). 4T1, DB-7, and MET-1 mouse cell lines were cultured in RPMI1640 supplemented with 10% FBS and insulin. MCF-10A cells were cultured in DMEM/F12 medium plus 5% horse serum, cholera toxin (100 ng/ml), EGF (20 ng/ml), insulin (10 μg/ml), and hydrocortisone (0.5 mg/ml). Cells were grown at 37 °C in 5% $CO_2$ incubators. Cells were obtained from ATCC, except those indicated below. MDA-MD231-derived LM2 and 4T1 cells were a kind gift from Dr. Joan Massague (Sloan Kettering Institute) and Dr. Haifa Shen (Houston Methodist Research Institute), respectively. DB-7 and MET-1 were described previously[46]. The TNBC cancer cell lines were recently authenticated by ATCC by using STR profiling. Cell lines were regularly tested being negative for mycoplasma.

**Chemicals**. Sources for chemicals are as follows: XY018 and GSK805 (purity > 99%) were synthesized by WuXi AppTec. Other chemicals (purity > 98%) are from Sigma and Selleck unless indicated otherwise.

**Organoid culture**. Organoids were cultured from PDX xenografts when the tumor size reached ~500 mm³. Briefly, dissected tumors were finely minced and transferred to a 50-ml conical tube, including a digestion mix consisting of serum-free DMEM/F-12 medium (Gibco) and 1 mg/ml collagenase IV (Sigma), and incubated for 1 h at 37 °C. Isolated organoids were mixed with 50 μl of Matrigel (BD Biosciences) and seeded in 24-well plates (Greiner bio-one). The culture medium contains phenol red-free DMEM/F-12 with penicillin/streptomycin/glutamine (100 mg/ml), primocin (50 mg/ml), Hepes (10 mM), B27 supplement (1 ×), A83-01 (500 nM), Y-27632 (5 μM), R-Spondin 3 (250 ng/ml), neuregulin 1 (5 nM), FGF 7 (5 ng/ml), FGF 10 (20 ng/ml), EGF (ng/ml), SB202190 (500 nM), N-acetylcysteine (1.25 mM), and nicotinamide (5 mM). One milliliter of supplemented culture medium was added per well, and organoids were maintained in a 37 °C humidified atmosphere under 5% $CO_2$.

**Mouse models and treatments**. NSG (JAX stock #005557) mice were purchased from the Jackson Laboratory. SCID C. B –17 mice or Balb/c nu/nu athymic mice were purchased from Envigo. Mice were housed under standard conditions with free access to food and water, under a 12-h light/12-h dark cycle in a temperature-controlled environment. Mice were fed a standard rodent chow diet (Envigo Teklad 2918). For cell line-derived xenografts, cells ($2 \times 10^6$ for each of the human cancer cells or $3 \times 10^4$ mouse 4T1) were mixed with Matrigel as 50% suspension. Then the cells were injected, in a volume of 0.1 ml, bilaterally into inguinal mammary glands of 4–6-week-old female C.B-17-SCID mice. For the patient-derived xenograft, PDX-1079 (JAX ID: TM01079), PDX-9327 (JAX ID: J000099327) and PDX-1173 (JAX ID: J000101173) were purchased from the Jackson Laboratory. PDXs were propagated by inserting ~2 mm³ into inguinal mammary glands of 4–6-week-old female C.B-17-SCID mice. Animal group size of six or more was estimated to have a high statistic power, based on power calculation and previous studies involving the same xenograft models. The concentrations of drug and the routes of drug administration are indicated in each figure. When the tumor volumes reached the indicated volume, mice were randomized and then administered with 100 μl of vehicle (with a formulation of 15% Cremophor EL, Calbiochem, 82.5% PBS, and 2.5% DMSO), statins (in PBS), or RORγ antagonists (in a formulation of 15% Cremophor EL, 82.5% PBS, and 2.5% DMSO), or their combinations (in their respective solvent). Tumor volumes were monitored by using calipers with volume calculated by using the equation: $\pi/6$ (length × width²). Body weight and survival during the course of the study was also monitored. At the end of the studies, mice

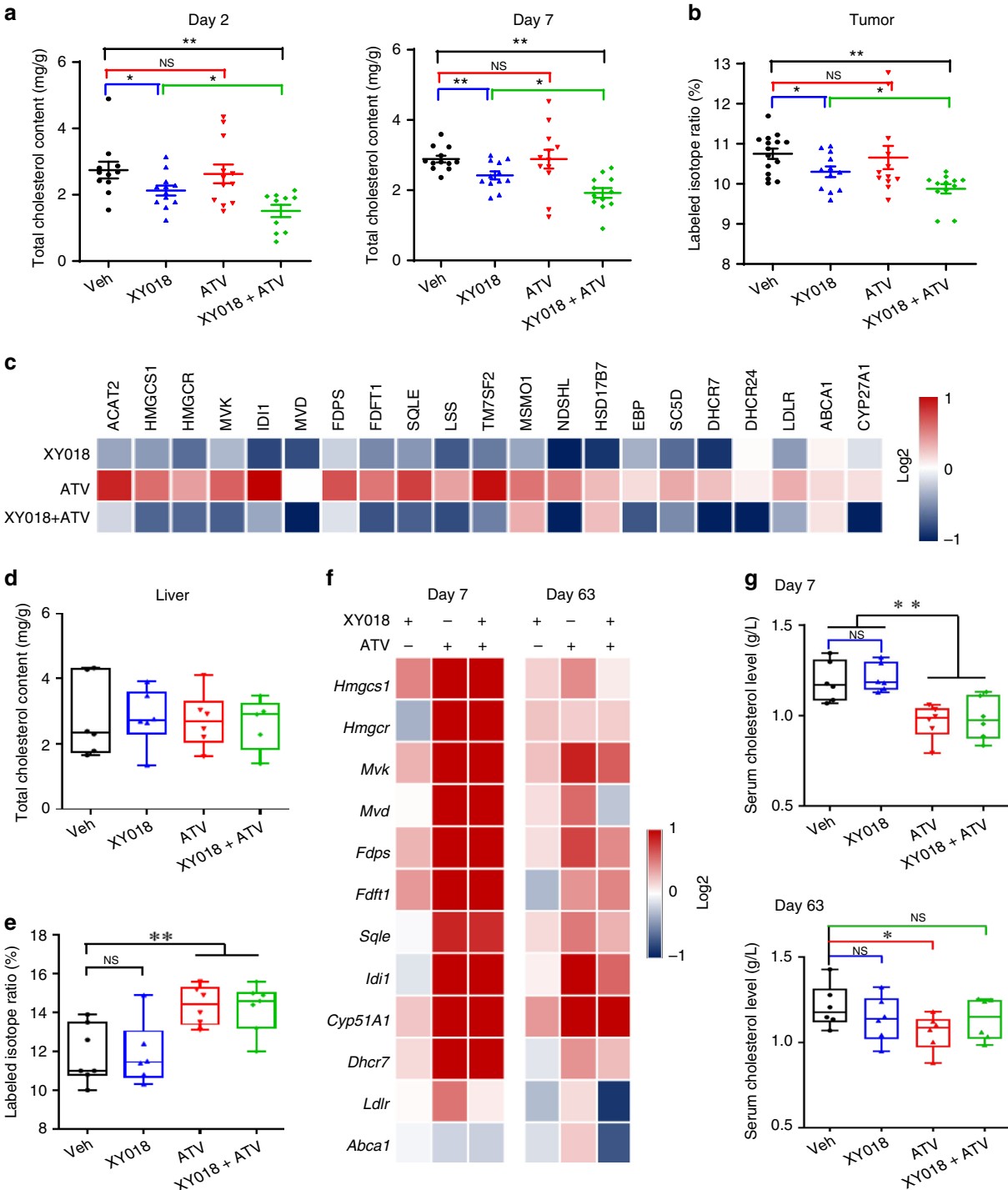

**Fig. 7** RORγ inhibitor reduces TNBC tumor cholesterol biosynthesis in vivo. **a** Total tumor cholesterol content from HCC70 tumor-bearing mice with indicated treatments for 2 or 7 days was analyzed after organic extraction. $n = 12–14$ tumors per group. **b** Labeled isotope ratio indicating tumor cholesterol biosynthesis rate was measured with GC–MS quantifying deuterated hydrogen incorporation into the newly synthesized cholesterol molecules. HCC70 tumor-carrying mice were treated with XY018, ATV, or both as indicated, then dosed with deuterium oxide for 2 h before tissues were collected. $n = 7$ mice per group. **c** Heat map display of fold changes (in log2) of cholesterol-biosynthesis pathway gene mRNA analyzed by qRT-PCR in tumors from HCC70 tumor-bearing mice with indicated treatments for 7 days. $n = 7$, the experiments were repeated three times. **d** Total liver cholesterol content from mice in (**a**) with indicated treatments for 7 days. $n = 7$. **e** Labeled isotope ratio of liver cholesterol biosynthesis rate in the mice as in (**b**). $n = 7$. **f** Heat map display of fold changes (in log2) of cholesterol-biosynthesis pathway gene mRNA analyzed by qRT-PCR in livers from TNBC tumor-bearing mice with indicated treatments for 7 or 63 days. $n = 7$, the experiments were repeated three times. **g** Total cholesterol levels in serum from mice as in (**f**) were measured. From **a** to **g**, 20 mg/kg XY018 or 15 mg/kg ATV were used. Data are shown as mean ± s.e.m. Student's $t$ test. $*p < 0.05$, $**p < 0.01$, NS not significant

were killed, and tumors were dissected and weighed. In addition, the organs and whole blood were harvested.

For effects on lung metastasis from orthotopic 4T1 murine tumors, female Balb/c mice were inoculated with 4T1 cells ($3 \times 10^4$) into the inguinal mammary gland fat pad. Once the tumors reached 250 mm³ at the primary sites they were surgically removed. Mice were then treated daily with XY018 (20 mg/kg, i.p.) or vehicle for 4 weeks. For histological analysis, mice were euthanized at the end of the treatments, and tumor nodules in the lungs were analyzed by H&E staining for morphology.

For effects on lung colonization, $2 \times 10^5$ luciferase-expressing MDA-MB231-LM2 (4175) cells were injected into each nude mouse via the tail. The mice were treated daily with vehicle or 20 mg/kg XY018, i.p. for 4 weeks. Tumor growth in the lungs was monitored for bioluminescence with a Xenogen IVIS-200 system.

The animal procedures were approved by the Institutional Animal Care and Use Committee (IACUC) of the University of California, Davis.

**Cell viability, apoptosis, and colony formation**. For cell viability, cells were seeded in 96-well plates at 1500–2500 cells per well in a total volume of 100 µl of media. After 4 days of incubation, Cell-Titer Glo reagents (Promega) were added, and luminescence was measured on GLOMAX microplate luminometer (Promega), according to the manufacturer's instructions. All experimental points were set up as sextuplicate as biological replication, and the entire experiments were repeated three times. The data are presented as percentage of viable cells with vehicle-treated cells set as 100. The estimated in vitro IC$_{50}$ values were calculated by using GraphPad Prism 8 software. Caspase-3/7 activity was measured by using a luminescent caspase-Glo 3/7 assay kit (Promega Corporation, Madison, USA), following the manufacturer's instructions. For cell growth, cells were seeded in 6-well plates at $2 \times 10^5$ per well and treated as indicated. Total viable cell numbers were counted with a Coulter cell counter. For colony formation, 500–1000 cells were seeded in a well of 6-well or 10-cm plates and cultured for 14 days with the medium changing every 3 days. When the cell clone grew visible, the medium was removed, and the cells were fixed with 10% formalin for 10 min. Then the plates were washed with PBS two times, and the cell colonies were stained with 0.2% crystal violet (in 10% formalin) for 15 min. The numbers of cell colonies were counted after washing 5 times by PBS. The above assays were performed in triplicates, and the entire experiments were repeated three times.

**Organoid viability**. For organoid viability, organoids were seeded in 96-well plates at 300–500 organoids in 10 µl of Matrigel per well in a total volume of 100 µl of media. Serially diluted compounds in 100 µl of media were added to the cells 12 h later. After 4 days of incubation, Cell-Titer Glo reagents (Promega) were added, and luminescence was measured. All experimental points were set up as sextuplicate as biological replication. The data are presented as percentage of viable cells with vehicle-treated cells set as 100.

After 6 days of incubation, medium was carefully aspirated and 100 µl of live/dead reagents (Thermofisher Scientific) was added followed by 30 min of incubation at room temperature. Fluorescence microscope was used to capture images of calcein AM (494/517 nm) to represent the live cells, of ethidium bromide homodimer-1 (528/617 nm) to identify the dead cells. The above assays were performed in triplicates. The entire experiments were repeated three times.

**qRT-PCR and western blotting analysis**. Total RNA was isolated from cells or xenograft tumors or mouse livers. The cDNA was prepared, amplified, and measured in the presence of SYBR as previously described with modification[27]. Briefly, the fluorescent values were collected, and a melting-curve analysis was performed. *GAPDH* was used as the internal reference to normalize the relative level of each transcript. Fold difference was calculated. The experiments were performed at least three times with data presented as mean values ± s.d. The primers are shown in Supplementary Table 2.

Cell lysates were analyzed by immunoblotting with antibodies specifically recognizing indicated proteins. The antibodies used are shown in Supplementary Table 3. The source data of the uncropped immunoblots are provided in the Source Data file.

**siRNA transfection and lentivirus production**. siRNAs for gene knockdown were purchased from Dharmacon. The siRNA target sequences for RORC were published previously[27]. siRNA against SREBP2 was purchased from Santa Cruz Biotechnology (sc-36559). Transfections were performed with OptiMEM (Invitrogen) and Dharmafectin#1 (Dharmacon) following the manufacturer's instruction. For RORγ overexpression, human RORγ cDNA in pLX304 (DNASU) was amplified and cloned into a modified pLX304 vector with a V5 tag at the receptor N terminus. Lentiviral particles were produced in 293T cells after co-transfection of the above lentivirus vectors, psPAX2 and pMD2.G in 10-cm dishes.

**CRISPR/Cas9 sgRNA design and lentivirus infection**. sgRNAs were designed by using the MIT CRISPR design software (http://crispr.mit.edu). Oligos corresponding to the sgRNAs were synthesized and cloned into lentiCRISPR v2 vectors following lentiCRISPRv2 and lentiGuide oligo-cloning protocol (Addgene, plasmid#52961). The sgRNA sequences are as follows: GFP,

GGGCGAGGAGCTGTTCACCG; sgRNA-1, GATACCCTCACCTACACCTT; sgRNA-2, GTGGGGCTGTCCAAGTGACC. Lentiviral particles were produced in 293T cells as in our previous study[27]. TNBC cancer cells were plated at $2 \times 10^5$ cells per well in 6-well plates. Sixteen hours later, 1 ml of virus-containing supernatant with 10 ng of polybrene was added to the cells. After 4–6 h, the medium was changed to regular medium and cultured for another 72 h before harvesting for cell number and protein expression analysis.

**RNA-seq alignment and analysis**. HCC70 or MDA-MB468 cells were treated with vehicle, the antagonists of XY018 (1.25 or 2.5 µM) or GSK805 (2.5 µM), ATV (1.25 µM), or the combination of antagonists and ATV for 24 h before RNA extraction. RNA-seq libraries from 1 µg of total RNA were prepared by using Illumina Tru-Seq RNA Sample Prep Kit, according to the manufacturer's instructions. Libraries were validated with an Agilent Bioanalyzer (Agilent Technologies, Palo Alto, CA). Sequencing was performed on an Illumina HiSeq 2000 sequencer at BGI Tech (Hong Kong). The FASTQ-formatted sequence data were analyzed by using a standard BWA–Bowtie–Cufflinks workflow. Sequence reads were mapped to GRCh37/hg19 assembly with BWA and Biotie software. The Cufflinks package was used for transcript assembly, quantification of normalized gene and isoform expression, and analysis of different expression. Gene Set Enrichment Analysis (GSEA v.3.0) was applied to rank genes based on the shrunken limma log2 fold changes. The GSEA tool was used in "pre-ranked" model with default parameters.

**Measurement of cholesterol contents in tissues and cells**. Cells or xenograft tumor tissues were washed three times with cold PBS and subjected to extraction with organic solvents (7:11:0.1, chloroform/isopropanol/Triton X-100). Total cholesterol levels were measured by using Amplex™ Red Cholesterol Assay Kit (Thermofisher Scientific) and normalized to protein concentrations. Total cholesterol in serum was measured by an enzymatic colorimetric method by using a commercial kit (Wako) according to the manufacturer's instructions. All experimental points were set up as triplicate as biological replication. The entire experiments were repeated three times. The cohort of breast cancer tumor tissues was previously described[47]. Their cholesterol contents were analyzed by using gas chromatography followed by time-of-flight mass spectrometry (GC–TOFMS) as described before[48]. Raw GC–TOFMS data were preprocessed by ChromaTOF version 2.32 for baseline subtraction, deconvolution, and peak detection. Specifically, 3-s peak width, baseline subtraction just above the noise level, and automatic mass spectral deconvolution and peak detection at signal/noise levels of 5:1 throughout the chromatogram were used. Binbase algorithm was used for annotation and relative quantification of cholesterol with the following settings: validity of the chromatogram 107 counts/s, unbiased retention index marker detection, MS similarity > 800, retention index calculation by fifth-order polynomial regression, retention index window 2000 units, and validation of unique ions and apex masses[49,50].

**Analysis of cholesterol-biosynthesis rate in tumor and liver**. SCID mice bearing HCC70 tumors were treated with indicated drugs for 2 days and then injected with 99% deuterium oxide (Cambridge Isotope Laboratories, 23.3 mg/g; i.p.). After 2 h, the mice were terminated under anesthesia; the liver and tumor were excised. Samples (10 mg) were extracted with 1 mL of extraction solvent, 3:3:2 acetonitrile, isopropanol, and water. The extraction solvent was pre-cooled at −20 °C. Samples were homogenized by using Genogrinder at $500 \times g$ for 30 s, shaken for 5 min at 4 °C, and centrifuged for 2 min at $20,000 \times g$. Five-hundred microliters of supernatant was dried overnight. Ten microliters of methoxyamine hydrochloride was added to each dried sample followed by shaking at 30 °C for 1.5 h. Ninety-one microliters of N-tert-Butyldimethylsilyl-N-methyltrifluoroacetamide (MTBSTFA, Sigma-Aldrich, St. Louis, MO) was added to each sample followed by shaking at 70 °C for 60 min for tert butyldimethylsilylation. Agilent 7200 GC-accurate-mass QTOF (Agilent, Santa Clara, CA) was used for data acquisition. Agilent J&W DB-5ms Ultra Inert column was used (Agilent, Santa Clara, CA). The column was held at 60 °C for 0.5 min, ramped to 325 °C at 10 °C/min, and held at 325 °C for 10 min. Raw data were processed by using Agilent Mass Hunter Quantitative Analysis software for QTOF (B. 07.00).

**Co-immunoprecipitation (Co-IP)**. Cells were washed three times and then lysed with lysis buffer (10 mM HEPES, pH 7.9, 10 mM KCl, 0.1 mM EDTA, 0.4% NP-40, and protease inhibitor cocktail) for 30 min at 4 °C. The homogenates were centrifuged for 30 s at $15,000 \times g$ at 4 °C. The supernatant was removed. The pellets were lysed in extraction buffer (20 mM HEPES, pH 7.9, 0.4 M NaCl, 1 mM EDTA, and protease inhibitor cocktail) for 15 min for nuclear extract collection. Magnetic beads (Thermofisher Scientific) were incubated with indicated antibodies at 4 °C for 2 h. Five percent of the nuclear extracts were harvested for western blot analysis as inputs. The remaining cell lysates were incubated with the pre-treated beads overnight at 4 °C. The immunoprecipitation beads were washed with wash buffer (50 mM Tris-HCl, pH7.5, 200 mM NaCl, 5 mM EDTA, and 1% Triton X-100) five times, followed by western blotting analysis. The source data of the uncropped immunoblots are provided in the Source Data file.

**Reporter constructs and reporter-gene assay**. Transient transfection and reporter-gene assays were performed as previously described[27], with the following modifications. For reporter-gene assays of cholesterol-biosynthesis genes, pGL3-MVK-wt and pGL3-HMGCS1-wt were constructed by inserting the DNA fragments of the MVK promoter from −787 to + 279 or the HMGCS1 promoter from −1288 to + 70 into pGL3-basic luciferase reporter vector. The mutant form pGL3-MVK-mu of pGL3-MVK-wt contains sequences mutated from ATTGGTCAC to ATTAACGTC. The mutant form pGL3-HMGCS1-wt contains sequences mutated from TCTGACCA to TCCCGTTA. Cells (HEK293T) were co-transfected with pLX304-RORγ or pcDNA3.1-SREBP2 (Addgene, plasmid#26807) with wild type or mutant forms of MVK or HMGCS1 promoter reporter constructs. The renilla plasmid was co-transfected for normalization. After 12 h of incubation, cells were treated with vehicle or RORγ antagonists as indicated, for another 24 h. The luciferase was then analyzed with a Dual-Luciferase Assay system (Promega) on a luminometer according to the manufactuer's instruction. All transfections were performed in sextuplicate, and each experiment was repeated at least three times.

**ChIP-qPCR, ChIP-seq, and data analysis**. Briefly, HCC70 cells were treated with vehicle or XY018 (2.5 μM) for 24 h before they were subjected to cross-linking in 1% formaldehyde for 5 min, followed by quenching with glycine for 5 min on ice. Cells were pelleted by centrifugation and resuspended in lysis buffer (50 mM HEPES, pH 8.0, 140 mM NaCl, 1 mM EDTA, 10% glycerol, 0.5% NP-40, and 0.25% Triton X-100). The pellets were then resuspended in washing buffer (10 mM Tris, pH 8.0, 1 mM EDTA, 0.5 mM EGTA, and 200 mM NaCl), washed, and resuspended in shearing buffer (0.1% SDS, 1 mM EDTA, pH 8, and 10 mM Tris-HCl, pH 8) before sonication by using Covaris E220 following the manufacturer's instruction. Chromatin fragments were precipitated by using specific antibodies and Protein G beads, washed, and treated with Proteinase K and RNAse A. Purified ChIP DNA was then used for ChIP-qPCR analysis and library generation.

The antibodies used for the ChIP-qPCR assay are SREBP2 (Cayman, 10007663); RNAPII (Santa Cruz; sc-899); RNAPII-S5P (Active Motif; #39749); RNAPII-S2P (Active Motif; #61083); H3K27ac (Abcam; ab4729); p300 (Abcam; ab10485); IgG (Santa Cruz; sc-2027). ChIPs were performed with each experimental point in triplicate, and each experiment was repeated three times. The primers are shown in Supplementary Table 2.

Libraries were quantified with the Bioanalyzer 2100 (Agilent) and sequenced in single-end 50-bp mode on the Illumina HiSeq 2000 Sequencer (BGI, Hong Kong). Antibodies used were against SREBP2 (Cayman, #10007663) and H3K27ac (Abcam; ab4729). Anti-RORγ rabbit serum was described previously[27]. The specificity data for the anti-RORγ antibody used in ChIP-seq are shown in the Source Data.

Fastq files from ChIP-seq were processed by the pipeline of AQUAS Transcription Factor and Histone (https://github.com/kundajelab/chipseq_pipeline). Briefly, sequencing tags were mapped against the Homo sapiens (human) reference genome (hg19) by using BWA 0.7.15[51]. Uniquely mapped tags after filtering and deduping were used for peak calling by model-based analysis for ChIP-Seq (MACS; 2.1.0) to identify regions of enrichment over background. Normalized genome-wide signal-coverage tracks from raw-read alignment files were built by MACS2, UCSC tools (bedGraphToBigWig/bedClip; http://hgdownload.cse.ucsc.edu/admin/exe/linux.x86_64/), and bedTools (https://github.com/arq5x/bedtools2). Visualization of the ChIP-seq signal at enriched genomic regions (avgprofile and heatmap) was achieved by using deepTools (https://deeptools.readthedocs.io/en/develop/index.html). Peak-associated genes were identified by using the annotatePeaks function of HOMER (http://homer.ucsd.edu/homer/index.html). Each of the RORγ or SREBP2-binding sites were assigned to the nearest gene. Further annotation information includes whether a peak is in the TSS (transcription start site, from −1 kb to + 100 bp), TTS (transcription termination site, from −100 bp to + 1 kb), Exon (Coding), 5′ UTR Exon, 3′ UTR Exon, Intronic, or Intergenic. Venn diagrams of enriched genomic regions and associated genes were generated by Intervene[52]. GO analysis was performed by using ClueGO[53] and BINGO[54].

**Motif analysis**. DNA segments from SREBP2-binding site of vehicle or XY018 (2.5 μM) treatment were extracted. The segments were used for motif analysis by homer script findMotifGenome.pl with argument "hg19-p32" to detect enrichment of de novo and known TF motif. Homer results were further manually analyzed. Those motifs with high background enrichment and possible false positives were removed.

**Bioinformatic analyses with data from clinical tumors**. METABRIC data sets for 1459 ER+ and 313 TNBC samples were downloaded from cBioPortal website at http://www.cbioportal.org/study?id=brca_metabric#summary. The data were then Log2 transformed and quantile normalized before further analysis. For each of the 10 metabolic pathways and cell cycle/proliferation pathway, a gene set was manually compiled. An "activity score" was represented by the first-principal component of the gene set for each ER+ or TNBC samples. Principal-component analysis (PCA) was carried out with R 'COMPADRE' package[55]. Differences in pathway activity between ER+ and TNBC groups were calculated by using two-sided Student's t tests. After PCA transformation, the samples were visualized

according to pathway activity score by using "gplots" R packages. Based on the pathway activity score and the gene profile across the samples, the Pearson correlation metric was computed between each gene (i.e., RORC) and each metabolic pathway by using the "cor" function in R.

The prognostic value of RORC gene characteristic with survival was determined by Kaplan–Meier analysis by using KM-plotter online software (http://kmplot.com/analysis/).

Kaplan–Meier estimates of the relationship between gene expression and distant metastasis-free survival were calculated by setting the software to look for the optimal cutoff for separation of patients into high- and low-expressing groups. The hazard ratio, log-rank P value, and number of patients in each group are shown on the KM plot for each gene.

**Morphology**. Sections (5 μm, 10 sections per sample) of liver or tumor tissues from tumor-bearing mice with or without treatment, were fixed in 10% phosphate-buffered formalin, stained with hematoxylin/eosin Y, and visualized by using a light microscope with a 20 × objective.

**Serum analysis**. Whole blood was collected via cardiac draw in Sarstedt 100-μL K3E EDTA tubes. The collected blood was immediately analyzed for complete blood count by using HemaVet 950FS (Drew Scientific). All biochemical serum evaluations used to investigate organ functions were performed at the same time to minimize analytical variability and determined on a Roche Integra 400 Plus analyzer (Roche Diagnostics).

**Statistical analysis**. Statistical analyses were performed by GraphPad Prism software 8.0. All statistical details of experiments are included in the figure legends or the specific Methods section.

**Reporting summary**. Further information on research design is available in the Nature Research Reporting Summary linked to this article.

## Data availability

All data supporting the findings of this study are available within the article and Supplementary Information, or from the corresponding author upon request. The source data underlying Figs. 3e, 4i, 5j and Supplementary Figs. 2e, f, h, k, 3c, d, g, 7a are provided as a Source Data file. All RNA-seq and ChIP-seq data generated in this study are deposited in the Gene Expression Omnibus (GEO) database under the accession numbers "GSE131856" and "GSE126380". A reporting summary for this article is available as a Supplementary Information file.

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

## Acknowledgements

We thank Dr. Peter Tontonoz for his valuable discussion and inputs. We thank Drs. Joan Massague and Haifa Shen for their kind gift of MB231-LM2 and 4T1 tumor cells, respectively. We also thank members of the Genomics Shared Resources of UC Davis Cancer Center for their technical help and Christopher Chen for his help in editing. This work was supported in part by grants from the NIH (R01CA224900 and R01CA206222 to H-W.C.). The UCDCCC Genomics Shared Resource is funded by the UC Davis Comprehensive Cancer Center Support Grant (CCSG) awarded by the National Cancer Institute (NCI P30CA093373).

## Author contributions

Conceptualization, D.C., J.W. and H.W.C.; Methodology, D.C., J.W., B.G., F.W., J.X.Z., J.X., A.D.B. and O.F.; Investigation, D.C., J.W., B.G., J.L., F.W., J.X.Z., J.X., Y.J., H.Z. and Z.H.; Writing—Original Draft, D.C. and H.W.C.; Writing—Review and Editing, D.C., H.W.C., P.N.L., H.J.K., R.Z. and R.M.E.; Funding Acquisition, O.F. and H.W.C.; Resources, A.D.B., P.N.L., R.J.B., J.J.L., X.C., K.S.L., R.Z., R.M.E. and K.F.T.; Supervision, H.W.C.

## Competing interests

The authors declare no competing interests.
