## [Peer Review File · Nature Communications]

Reviewers' Comments:

Reviewer #1:

Remarks to the Author:

The manuscript by Cai et al. reports an interesting role of ROR γ in controlling cholesterol metabolism in triple-negative breast cancer (TNBC). They found that TNBCs have elevated cholesterol biosynthesis (CB) pathway in TNBC compared to ER+ breast cancer. Using various antagonists and agonists, they found that the CB in TNBC was controlled by ROR γ , but not in ER+ cancer cells. ROR γ is important for the TNBC cancer cell proliferation/survival in vitro and for tumor growth in vivo. Antagonists of ROR γ acted synergistically with statins in suppressing TNBC tumor growth in cell line and PDX models. The role of ROR γ in regulating CB has not been reported before. The authors further investigated how ROR γ acted mechanistically in this process. Interestingly, they found that ROR γ was needed for the chromatin binding and hence the action of the master regulator of CB, SREBP2. However, this mechanism only occurs in TNBC, but not in ER+ cancer cells. In keeping with the specific role of ROR γ in CB in TNBC, ROR γ antagonists only affect CB in tumor cells but not the hosts.

Overall, this is an interesting study uncovering a new and actionable signaling pathway in TNBC, which still lacks targeted therapy. Most of the data presented are robust and of good quality, although there are several points that need to be addressed before publication.

1. A key conclusion is that ROR γ acts by enabling SREBP2 chromatin binding and activating target gene transcription. They have shown this by ChIP-seq, co-IP, and luciferase reporter experiments. However, the result in Fig. 3h is contradictory to this conclusion. If ROR γ only acts through activating SREBP2, SREBP2 knockdown should inhibit the upregulation of CB genes by ROR γ . However, Fig. 3h shows SREBP2 has no effect at all. If this is indeed the case, one has to conclude that ROR γ can induce CB independent of SREBP2. The authors need to address this contradiction.
2. In Fig. 4d, the ROR γ inhibitor XY018 had no effect on ROR γ chromatin binding, but significantly affected SREBP2 binding. This result needs clarification. Is it known that XY018 does not affect ROR γ chromatin binding, rather affecting its association with other proteins? What about other inhibitors? Normally the nuclear receptor inhibitors should affect their binding to DNA.
3. What are the expression levels of ROR γ in all the TNBC and ER+ breast cancer cell lines? Could it be the fact that MCF7 is not sensitive to ROR γ antagonist is due to MCF7 express very low ROR γ ? If so, can ROR γ overexpression in MCF7 have any effect in the CB pathway?
4. Fig. 4h claims ROR γ and SREBP2 acts synergistically. The effects are more like additive rather than synergistic.
5. The authors claim that ROR γ inhibitor plus statin can achieve sustain tumor regression (fig. 6d) in both the abstract and the main text. This is not correct. The tumor stopped grow, however the size did not shrink. This is stasis rather than regression.

Reviewer #2:

Remarks to the Author:

In this very elegant paper, the authors studied metabolic deregulations that could discriminate TNBC from ER+BC using the METABRIC database. They found an upregulation of cholesterol biosynthesis in TNBC compared to ER+ suggesting this difference could be exploited for the development of new therapeutic strategies against TNBC. They next developed a strategy based on the comparison of well-studied TNBC and ER+ BC cell lines phenotypes regarding cholesterol biosynthesis (CB) and identified ROR γ as a possible target for TNBC and confirmed it in vitro and in vivo using genetic and pharmacological approaches.

This is a very interesting study that offers a promising therapeutic perspective for TNBC treatments, which remains to date an unmet medical need.

Questions:

1. While the authors are suggesting in the introduction that cholesterol metabolism was linked to the activation of mitogenic pathway mainly driven by cell surface signalling cholesterol per se is not a mitogenic compound. This is a bit confusing to me.
2. the authors must check the expression of genes involved cholesterol metabolism the could produce tumor promoters such as cholesteryl esters and 6oxo-cholestan-diol
3. The methodology used for cholesterol quantification is not clear. is it an absolute or a relative quantification method? I cannot found details in the mat & meth section and in the articles they refered to.
4. There is cell type modulation of CB enzyme according cell lines and treatments. The determination of the sterol profile (cholesterol precursors such as lanosterol, 7dehydrocholesterol, desmosterol, ...) would be very informative.
5. What is the expression level of NR proteins and of RORgamma in the tested cancer cells?
6. on page 5 last paragraph: the genes involved in Cholesterol biosynthesis used for their correlations must be specified
7. on page 6: the authors are measuring apoptosis markers to determine the impact of RORgamma on cancer cell growth. Do caspase inhibitors restored cancer cell proliferation?
8. same page what is the protein expression level of RORgamma in cells overexpressing ROR gamma compared to untransfected cells?
9. page 6 and 7: to me the authors evidenced that RORgamma is more a regulator of apoptosis than of cell proliferation.
10. page 8-9: « the steroid/cortisol synthesis » program : please could give more details since cortisol metabolic enzymes HSD11B1 and B2 are involved in oxysterol metabolism including an oxysterol with tumor promoter properties (Voisin et al, PNAS, 2017 ; Beck et al, J Steroid Biochem Mol Biol, 2019).

Reviewer #3:

Remarks to the Author:

Summary:

The current manuscript implicates ROR γ as a critical regulator of cholesterol biosynthesis within triple negative breast cancer cells. Previous studies have shown that various cholesterol precursors/metabolites have the capacity to bind to and modulate the activity of the RORs, suggesting that the RORs had roles in regulating cholesterol homeostasis. Interestingly however, the current study has found that this regulation is rather modest except within TNBC cells, perhaps because this subtype is often characterized by increased cholesterol biosynthesis.

Key findings include: (1) ROR γ is dominant over SREBP2 in terms of regulating cholesterol biosynthesis. This is likely due to its ability to directly bind to and facilitate recruitment of SREBP2. This is a novel finding in terms of cholesterol homeostasis. The finding that knockdown of SREBP2 had no effect on the cholesterol synthesis program in cells overexpressing ROR γ , and is an emerging theme indicating that SREBP2 is not always the master regulator of cholesterol biosynthesis. (2) TNBC cells are particularly sensitive to manipulations of ROR γ , especially compared to ER+ models. (3) ROR γ antagonists reduce the growth of TNBC xenografts and a syngeneic 4T1 graft. Metastasis of 4T1 and MDA MB 231 models is also reduced. Combination therapy of ROR γ antagonists with statins further reduced tumor growth (in fact, pretty much inhibiting growth altogether).

This is a timely study, especially given recent interest in the impact of cholesterol and fatty acid metabolism on cancer. The experiments are thorough and include several complementary

approaches, reveal novel mechanisms and establish ROR γ as a potential therapeutic target. The figures are very well organized, and the text is well written. There are a few concerns that should be considered prior to acceptance for publication.

Major:

1) Tumor studies

a. As far as this reviewer can tell, treatment was initiated very shortly after the tumor graft. Thus, the effects could be due to influence of tumor establishment rather than subsequent outgrowth (and thus does not reflect a clinically relevant scenario). This is purported as one reason many translational therapeutics fail in the clinic. Therefore, the authors should test the efficacy of ROR γ antagonists (+/- statins) after the tumor has already established (ie: a tumor size of $\sim 200\text{mm}^3$ for cell grafts 4T1 AND 468).

b. A direct comparison of ROR γ antagonist with standard of care chemotherapy should be made for at least one of the tumor models.

2) Language/claims: The authors make several claims which are somewhat of a stretch of truth.

a. Introduction: "ROR has not been implicated in control of tumor CB or CHOL homeostasis". ROR has been implicated in cholesterol biology (see its endogenous ligands), as well as tumor biology.

b. Abstract: "ROR functions as a master activator of the entire MVA-CB program". Based on the data in this manuscript, this claim is really only relevant to TNBC.

c. Results: "33 small-molecule modulators targeting SREBP2 and members of the NR family...."

Only one of these molecules targets SREBP2 (Fatostatin). There are other reported modulators of SREBP2 that are not included. These should be included, or at least the text be reworked to better reflect the selected molecules.

3) CHIP studies: The authors should demonstrate the specificity of their chosen antibodies against ROR and SREBP2.

4) Results: "...thus suggesting that ROR γ can act as a coactivator of SREBP2". The data also support alternative models – especially given the continued activity of ROR in the absence of SREBP2. This would suggest that ROR is independently regulating these genes.

5) The reduction in circulating cholesterol levels upon treatment with statins is somewhat surprising given that mice typically have very low circulating levels to start with, and several previous studies indicating that oral statins have little impact on circulating cholesterol levels in wildtype murine models on standard diets. The authors should address this potential discrepancy in the text provide further details of their chosen mouse models and diets (ie: cholesterol content and fat content).

6) Of the genes within the CHOL-biosynthesis pathway, which ones are driving the correlation with RORC expression (Fig. 1d).

Minor:

- The acronyms ROR, LXR etc. should be defined the first time they are used in the text.

- Figure legends lack experimental detail.

- The use of acronyms can be frustrating. This reviewer suggests spelling out cholesterol, cholesterol biosynthesis, mevalonate etc....

- Results "...SREBP2 binding sites was also....", should read "...SREBP2 binding sites were also...."

- Results "Having... through blocking the reprogramming." Reprogramming of what?

- Discussion – when describing effects neutrophils, Th17 and gd-T cells. Baek et al, Nat Com, 2017 should also be cited, as this paper directly describes influence of cholesterol/cholesterol metabolites on these cell types in the context of breast cancer.

- Discussion and citations of papers describing the purported endogenous ligands for the RORs should also be included.

- Figure 2e: "none-tnbc" should read "non-tnbc"

- Rather than the pictures of tumors/mice in Fig 6d and e, the authors should consider including the growth curves in Supplemental Fig 6c,j and K etc. It is much more impressive that the drug combinations work across several models.

Reviewer #4:

Remarks to the Author:

The study by Cai et al. identifies the nuclear receptor ROR γ as a novel master regulator of cholesterol metabolism and homeostasis in triple negative breast cancer. The authors go on to show that this protein is targetable with small molecule antagonists that selectively inhibit growth of human TNBC cells in vivo. This is a highly novel finding that has wide ramifications, including beyond breast cancer. Notably, the authors demonstrate that in TNBC cells, ROR γ is dominant over SREBP-2, a transcription factor well established as a master regulator of cholesterol metabolism. The findings identify important biochemical distinctions and dependences between TNBC and ER+ breast cancer. As a result, the study will have wide interest in the breast cancer community. However, the principal observation may be more fundamental and may have wide applicability in other cancers and other physiologic settings, likely generating interest much more widely. The study is comprehensively done, with multiple models, a wide range of experimental approaches, and an extensive data set in support of the conclusions of the study. Statistics appear satisfactory. There are some issues that could be addressed to strengthen the study further:

Major points:

1. Although human data are included, the treatment of human breast cancer data is rather superficial. RNA expression level of transcription factors is not a satisfactory indicator of activity. The RNA profiling and ChIP-seq data allow the development of an ROR γ activity signature that could be read out in human breast cancer RNA expression profiles and many other datasets. Because extensive human breast cancer datasets exist in the public domain, not having these data limits the translational impact of the study.
2. The exclusive focus on TNBC vs. ER+ cancer is appropriate, but because the master regulator discovery could potentially operate in many other scenarios, it would be helpful if there was some sense from data presented here that this mechanism operates elsewhere. This could be done by analyzing inferred ROR γ activity using a gene signature panel as mentioned above.
3. There should be a better description of the specificity, selectivity and potential limitations of the ROR γ antagonists, since non-selectivity would confound one of the principal conclusions of the study.
4. There is quite a bit of sloppiness in the text, particularly with respect to standard English.

Minor point:

The header label in figure 3e is mis-aligned.

Response to Referees

For your convenience, we highlight changes (mostly new data) in red in the revised manuscript.

Reviewer #1 (Remarks to the Author):

The manuscript by Cai et al. reports an interesting role of ROR in controlling cholesterol metabolism in triple-negative breast cancer (TNBC). They found that TNBCs have elevated cholesterol biosynthesis (CB) pathway in TNBC compared to ER+ breast cancer. Using various antagonists and agonists, they found that the CB in TNBC was controlled by ROR, but not in ER+ cancer cells. ROR is important for the TNBC cancer cell proliferation/survival in vitro and for tumor growth in vivo. Antagonists of ROR acted synergistically with statins in suppressing TNBC tumor growth in cell line and PDX models.

The role of ROR in regulating CB has not been reported before. The authors further investigated how ROR acted mechanistically in this process. Interestingly, they found that ROR was needed for the chromatin binding and hence the action of the master regulator of CB, SREBP2. However, this mechanism only occurs in TNBC, but not in ER+ cancer cells. In keeping with the specific role of ROR in CB in TNBC, ROR antagonists only affect CB in tumor cells but not the hosts.

Overall, this is an interesting study uncovering a new and actionable signaling pathway in TNBC, which still lacks targeted therapy. Most of the data presented are robust and of good quality, although there are several points that need to be addressed before publication.

We thank this reviewer for his/her many positive comments of our manuscript. We also very much appreciate the constructive suggestions.

1. A key conclusion is that ROR acts by enabling SREBP2 chromatin binding and activating target gene transcription. They have shown this by ChIP-seq, co-IP, and luciferase reporter experiments. However, the result in Fig. 3h is contradictory to this conclusion. If ROR only acts through activating SREBP2, SREBP2 knockdown should inhibit the upregulation of CB genes by ROR. However, Fig. 3h shows SREBP2 has no effect at all. If this is indeed the case, one has to conclude that ROR can induce CB independent of SREBP2. The authors need to address this contradiction.

Response

We thank the reviewer for pointing out an interpretation of data in Fig.3h that can be potentially contradictory to the conclusion that ROR function by enabling SREBP2 chromatin binding and activating target gene transcription. Here we would like to point out the specific experimental condition in Fig 3h where ROR γ protein is ectopically overexpressed. We believe that this ectopically overexpressed ROR γ can function independent of SREBP2. However, in the natural setting such as cancer cells or tumor tissues without ectopically overexpressed ROR γ , we believe that the main functional mode of ROR γ in CB pathway is through promoting SREBP2 chromatin binding, which are supported by data in Figures 3d, 3g, 4d, 4e and 4f where there're no ectopically overexpressed ROR γ . We now made clarification in the results description by indicating that cells are ectopically overexpressing ROR γ .

2. In Fig. 4d, the ROR inhibitor XY018 had no effect on ROR chromatin binding, but significantly affected SREBP2 binding. This result needs clarification. Is it known that XY018 does not affect ROR chromatin binding, rather affecting its association with other proteins? What about other inhibitors? Normally the nuclear receptor inhibitors should affect their binding to DNA.

Response

These are very interesting questions. First, it is not known whether XY018 affects ROR γ chromatin binding before this study. In this study our genome-wide analysis does not show significantly effect on ROR γ chromatin binding (Fig.4d). In terms of other ROR γ inhibitors it was reported that ROR γ inhibitors/antagonists can elicit distinct effects on ROR γ chromatin binding; some affect ROR γ chromatin binding while others don't¹. Therefore, the effects of nuclear receptors inhibitors on the receptor chromatin binding can be compound specific.

3. What are the expression levels of ROR in all the TNBC and ER+ breast cancer cell lines? Could it be the fact that MCF7 is not sensitive to ROR antagonist is due to MCF7 express very low ROR ? If so, can ROR overexpression in MCF7 have any effect in the CB pathway?

Response

To address the question, we now provide in supplementary Fig.2h the ROR γ protein expression in different breast cancer cells including TNBC and ER+ ones. The results show that ROR γ expression in MCF-7 is no less than most of the TNBC cells. Therefore, the lack of sensitivity to ROR γ antagonist by MCF-7 cells is not due to lack or low expression of ROR γ in the cells.

4. Fig. 4h claims ROR and SREBP2 acts synergistically. The effects are more like additive rather than synergistic.

Response

We now provide new data in Fig. 4h which shows that while SREBP2 or ROR γ alone can activate the reporter about 7 or 8 fold, their combination resulted in over 30 fold of transactivation of the MVK gene promoter. Therefore, the new data demonstrate that ROR γ and SREBP2 act synergistically.

5. The authors claim that ROR inhibitor plus statin can achieve sustain tumor regression (fig. 6d) in both the abstract and the main text. This is not correct. The tumor stopped grow, however the size did not shrink. This is stasis rather than regression.

Response

We now removed the word “sustain” from abstract and main text. Although we did not show the continued regression/tumor size decrease, we would like to respectfully point out that the combined treatment does cause tumor size reduction from initial 100mm³ to around 50 mm³ in Fig. 6d, which demonstrates that combined treatment induced tumor regression. We also showed that ROR γ inhibitor alone treatment can reduce tumor size from around 160 mm³ to around 60 mm³ over 4 weeks as shown in Fig.6a.

Reviewer #2 (Remarks to the Author):

In this very elegant paper, the authors studied metabolic deregulations that could discriminate TNBC from ER+BC using the METABRIC database. They found an upregulation of cholesterol biosynthesis in TNBC compared to ER+ suggesting this difference could be exploited for the development of new therapeutic strategies against TNBC. They next developed a strategy based on the comparison of well-studied TNBC and ER+ BC cell lines phenotypes regarding cholesterol biosynthesis (CB) and identified ROR γ as a possible target for TNBC and confirmed it in vitro and in vivo using genetic and pharmacological approaches.

This is a very interesting study that offers a promising therapeutic perspective for TNBC treatments, which remains to date an unmet medical need.

We thank this reviewer for his/her recognition of the significance of our study and his/her valuable comments.

Questions:

1. While the authors are suggesting in the introduction that cholesterol metabolism was linked to the activation of mitogenic pathway mainly driven by cell surface signalling cholesterol per se is not a mitogenic compound. This is a bit confusing to me.

Response

We apologize that we did not make this point clear in the introduction. We agree that cholesterol per se is not mitogenic compound. We now modified some of the sentences highlighted in red in the first paragraph of introduction. We also cited two additional references (#6, Voisin, M. *et al.* PNAS, 2017; Poirot, M *et al.* Biochimie,2018).

2. the authors must check the expression of genes involved cholesterol metabolism the could produce tumor promoters such as cholesteryl esters and 6oxo-cholestan-diol

Response

We agree with the comments that cholesterol metabolites such as cholesteryl esters and 6oxo-cholestan-diol can be important in tumor development. We have checked the expression of key genes including SOAT1/ACAT1, SOAT2, LCAT, H6PD, HSD11B1 and HSD11B2 in our RNA-seq data. However, we did not find any significant changes in response to treatments of ROR γ inhibitor or statins or their combination as shown below.

gene id	Symbol	Veh (FPKM)	XY018 1.25 μ M (FPKM)	ATV 1.25 μ M (FPKM)	XY018 + ATV (FPKM)
6646	SOAT1	12.44	14.99	11.09	11.86
8435	SOAT2	0	0	0	0
3931	LCAT	1.45	2.03	1.32	1.58
9563	H6PD	18.63	18.35	21.37	21.92
3290	HSD11B1	0	0	0	0.07
3291	HSD11B2	7.35	5.45	7.84	6.64

3. The methodology used for cholesterol quantification is not clear. is it an absolute or a relative quantification method? I cannot found details in the mat & meth section and in the articles they refered to.

Response

We apologize for a lack of clarity of cholesterol quantification method. We used the relative quantification method. We have clarified this in the Methods as below.

The cohort of breast cancer tumor tissues were previously described. Their CHOL contents were analyzed using gas chromatography followed by time of flight mass spectrometry (GC-TOFMS) as described before. Raw GC-TOFMS data was preprocessed by ChromaTOF version 2.32 for baseline subtraction, deconvolution and peak detection. Specifically, 3s peak width, baseline subtraction just above the noise level, and automatic mass spectral deconvolution and peak detection at signal/noise levels of 5:1 throughout the chromatogram were used. Binbase algorithm was used for annotation and relative quantification of cholesterol with the following settings: validity of chromatogram 10^7 counts/s, unbiased retention index marker detection, MS similarity > 800, retention index calculation by 5th order polynomial regression, retention index window 2000 units and validation of unique ions and apex masses^{2,3}

4. There is cell type modulation of CB enzyme according cell lines and treatments. The determination of the sterol profile (cholesterol precursors such as lanosterol, 7dehydrocholesterol, desmosterol, ...) would be very informative.

Response

We agree that it would be very informative if we could provide sterol profiles. However, we would like to respectfully indicate that this line of study is a part of our ongoing projects which involves multiple approaches. We hope that we will be able to report the results in the future.

5. What is the expression level of NR proteins and of RORgamma in the tested cancer cells?

Response

To address the question, we now provide in supplementary Fig.2h the western blotting data for ROR γ and other ROR γ -closely related NR proteins including ROR α , REV-ERB α and LXR α in different breast cancer cells including TNBC and ER+ ones.

6. on page 5 last paragraph: the genes involved in Cholesterol biosynthesis used for their correlations must be specified

Response

As suggested, we now added these cholesterol biosynthesis genes in supplementary Fig. 1d with positive correlation genes highlighted in red.

7. on page 6: the authors are measuring apoptosis markers to determine the impact of RORgamma on cancer cell growth. Do caspase inhibitors restored cancer cell proliferation?

Response

Yes, we have performed the experiment using a caspase inhibitor (Caspase-3/7 Inhibitor I, 218826, Calbiochem) as suggested. The results demonstrate that this caspase inhibitor can at least partially restore the proliferation of TNBC cells treated with ROR γ inhibitor as below.

[redacted]

8. same page what is the protein expression level of RORgamma in cells overexpressing ROR gamma compared to untransfected cells?

Response

We now added the ROR γ expression in the other two cells SUM159 and MDA-MB231 in addition to the data MDA-MB468 in previous version as shown in supplementary Fig. 2f.

9. page 6 and 7: to me the authors evidenced that RORgamma is more a regulator of apoptosis than of cell proliferation.

Response

We agree with your assessment that ROR γ is more a regulator of apoptosis than of cell proliferation. We now modified the sentence that “Together, these results strongly support the notion that ROR γ is a major driver of TNBC cell survival”.

10. page 8-9: « the steroid/cortisol synthesis » program: please could give more details since cortisol metabolic enzymes HSD11B1 and B2 are involved in oxysterol metabolism including an oxysterol with tumor promoter properties (Voisin et al, PNAS, 2017 ; Beck et al, J Steroid Biochem Mol Biol, 2019).

Response

As suggested, we now provide the gene list (supplementary Table 1) for each of the programs related to Fig. 4b. HSD11B2 is included in the steroid hormone biosynthesis program. Also, paper by Viosin *et al*, PNAS, 2017 is now cited in the introduction.

Reviewer #3 (Remarks to the Author):

Summary:

The current manuscript implicates ROR γ as a critical regulator of cholesterol biosynthesis within triple negative breast cancer cells. Previous studies have shown that various cholesterol precursors/metabolites have the capacity to bind to and modulate the activity of the RORs, suggesting that the RORs had roles in regulating cholesterol homeostasis. Interestingly however, the current study has found that this regulation is rather modest except within TNBC cells, perhaps because this subtype is often characterized by increased cholesterol biosynthesis.

Key findings include: (1) ROR γ is dominant over SREBP2 in terms of regulating cholesterol biosynthesis. This is likely due to its ability to directly bind to and facilitate recruitment of SREBP2. This is a novel finding in terms of cholesterol homeostasis. The finding that knockdown of SREBP2 had no effect on the cholesterol synthesis program in cells overexpressing ROR γ , and is an emerging theme indicating that SREBP2 is not always the master regulator of cholesterol biosynthesis. (2) TNBC cells are particularly sensitive to manipulations of ROR γ , especially compared to ER+ models. (3) ROR γ antagonists reduce the growth of TNBC xenografts and a syngeneic 4T1 graft. Metastasis of 4T1 and MDA MB 231 models is also reduced. Combination therapy of ROR γ antagonists with statins further reduced tumor growth (in fact, pretty much inhibiting growth altogether).

This is a timely study, especially given recent interest in the impact of cholesterol and fatty acid metabolism on cancer. The experiments are thorough and include several complementary approaches, reveal novel mechanisms and establish ROR γ as a potential therapeutic target. The figures are very well organized, and the text is well written. There are a few concerns that should be considered prior to acceptance for publication.

We really appreciate this reviewer's recognition of our study being timely, thorough with complementary approaches and novel.

Major:

1) Tumor studies

a. As far as this reviewer can tell, treatment was initiated very shortly after the tumor graft. Thus, the effects could be due to influence of tumor establishment rather than subsequent outgrowth (and thus does not reflect a clinically relevant scenario). This is purported as one reason many translational therapeutics fail in the clinic. Therefore, the authors should test the efficacy of ROR γ antagonists (+/- statins) after the tumor has already established (ie: a tumor size of $\sim 200\text{mm}^3$ for cell grafts 4T1 AND 468).

b. A direct comparison of ROR γ antagonist with standard of care chemotherapy should be made for at least one of the tumor models.

Response

We apologize that we did not make it very clearly when we started the treatment after tumor grafting. For the two PDX models presented in Fig. 6, the treatment did not start until over 5 weeks after the initial tumor grafting. We then started the treatment when the tumor size reached to between 100 to 200 mm^3 . Therefore, our treatment was not initiated very shortly after the tumor grafting.

For the 4T1 model, we were using a highly metastatic cell line. Tumors derived from this cell line can kill mice around 2 weeks after the initial grafting due to metastasis to lung and other essential organs. Similar phenotype has been reported⁴⁻⁶. This is why most of the studies start their treatment when the tumor size is less than 100 mm^3 . Nevertheless, as suggested we performed the treatment when the tumor size reached to around 200 mm^3 and found that ROR γ inhibitor is still effective in inhibition of the 4T1 tumor growth as shown below. However, due to the fact that many mice were dead starting around day 9 of treatment we had to end this experiment. Also as suggested we have performed comparison study for the efficacy of ROR γ antagonists with one of the chemo-drugs (Doxorubicin). We found that ROR γ inhibitor shows better tumor inhibition efficacy than that of Dox treatment as shown below. Since this is part of our ongoing study, we present the data here to the reviewers.

[Redacted]

Also as suggested, we performed the treatment of MDA-MB468 model after the tumors size reached to around 200 mm^3 and found that ROR γ antagonist when combined with atorvastatin can effectively stop tumor growth as shown in Fig. 6g.

2) *Language/claims: The authors make several claims which are somewhat of a stretch of truth.*

a. Introduction: “ROR has not been implicated in control of tumor CB or CHOL homeostasis”. ROR has been implicated in cholesterol biology (see its endogenous ligands), as well as tumor biology.

Response

From reading this particular comment, we feel that this reviewer was confused by our use of abbreviations for cholesterol biosynthesis namely CB. The reviewer may think we use CB for cancer biology. We apologize for this confusion. Moreover, we would like to point out that although cholesterol precursors or metabolites have been shown to bind to ROR γ and regulate its transcriptional activity particularly in Th17 cell differentiation⁷⁻⁹, so far ROR γ itself has not been clearly implicated in control of tumor cholesterol biosynthesis or homeostasis. Also, actually this sentence “ROR has not been implicated in control of tumor CB or CHOL homeostasis” is the 2nd paragraph of the Results. We now modified this sentence in the 2nd paragraph of Results.

b. Abstract: “ROR functions as a master activator of the entire MVA-CB program”. Based on the data in this manuscript, this claim is really only relevant to TNBC.

Response

Yes, we agree with the assessment that this claim is only relevant to TNBC. This TNBC-relevant notion is reflected in the Title “ROR γ is a targetable master regulator of cholesterol biosynthesis in a subtype of breast cancer”, also in the second sentence of the Abstract “Here we show that triple-negative breast cancer (TNBC) exhibits a hyper-activated cholesterol biosynthesis program that is strongly linked to nuclear receptor ROR γ , compared to estrogen receptor-positive breast cancer” and in the last sentence of the Abstract “Together, our studies uncover a previously unsuspected master regulator of cholesterol biosynthesis and an attractive target for TNBC”.

c. Results: “33 small-molecule modulators targeting SREBP2 and members of the NR family....” Only one of these molecules targets SREBP2 (Fatostatin). There are other reported modulators of SREBP2 that are not included. These should be included, or at least the text be reworked to better reflect the selected molecules.

Response

We now added data from another SREBP-2 targeting compound (PF-429242) and modified the sentence to “31 small molecules targeting NR family members and 2 compounds that target SREBP2 translocation regulators SCAP (fatostatin) or S1P (PF-429242)”. The results show that PF-429242 displayed similar effects on cholesterol metabolism gene expressions when compared to fatostatin as shown in Fig. 1c.

3) *ChIP studies: The authors should demonstrate the specificity of their chosen antibodies*

against ROR and SREBP2.

Response

As suggested, we now added the specificity data for anti-ROR γ antibody used in ChIP in the last page of Source Data. The specificity of anti-SREBP2 antibody we used has been validated by ENCODE consortium which can be accessed at the website (<https://www.encodeproject.org/antibodies/ENCAB000ALD/>). We also indicate this in the report summary.

4) Results: "...thus suggesting that ROR γ can act as a coactivator of SREBP2". The data also support alternative models – especially given the continued activity of ROR in the absence of SREBP2. This would suggest that ROR is independently regulating these genes.

Response

We agree with the assessment that ROR γ can independently regulate these genes under specific conditions. However, we would like to point out the specific experimental condition in Fig. 3h where ROR γ protein is ectopically overexpressed. We believe that this ectopically overexpressed ROR γ can function independent of SREBP2. However, in the natural setting such as cancer cells or tumor tissues without ectopically overexpressed ROR γ , we believe that the main functional mode of ROR γ in cholesterol biosynthesis pathway is through promoting SREBP2 chromatin binding, which are supported by data in Fig. 3d, 3g, 4d, 4e and 4f where there're no ectopically overexpressed ROR γ . We now made clarification in the results description by indicating that cells are ectopically overexpressing ROR γ .

5) The reduction in circulating cholesterol levels upon treatment with statins is somewhat surprising given that mice typically have very low circulating levels to start with, and several previous studies indicating that oral statins have little impact on circulating cholesterol levels in wildtype murine models on standard diets. The authors should address this potential discrepancy in the text provide further details of their chosen mouse models and diets (ie: cholesterol content and fat content).

Response

We understand that there are previous studies that may suggest that statins have low effects on circulating cholesterol level in mice. However, our results are consistent with a recent comprehensive study¹⁰. In this study multiple approaches were used to make comprehensive measurements. The data clearly demonstrate that oral atorvastatin or lovastatin significantly decreases blood cholesterol level in mice. Also, in another study oral simvastatin is able to significantly decrease blood cholesterol level in mice¹¹. In terms of diets used, we used the standard rodent chow diets as in the above-mentioned studies as described in Methods.

6) Of the genes within the CHOL-biosynthesis pathway, which ones are driving the correlation with RORC expression (Fig. 1d).

Response

As suggested, we now added these cholesterol biosynthesis genes in supplementary Fig. 1d with positive correlation genes highlighted in red.

Minor:

- *The acronyms ROR, LXR etc. should be defined the first time they are used in the text.*

Response

Corrected.

- *Figure legends lack experimental detail.*

Response

We now modified some of the figure legends to include more details.

- *The use of acronyms can be frustrating. This reviewer suggests spelling out cholesterol, cholesterol biosynthesis, mevalonate etc....*

Response

Modified.

- *Results "...SREBP2 binding sites was also....", should read "...SREBP2 binding sites were also...."*

Response

Corrected.

- *Results "Having... through blocking the reprogramming." Reprogramming of what?*

Response

Here we refer "the reprogramming" at the end of this sentence in the paragraph to "reprogramming of TNBC *de novo* cholesterol biosynthesis in vitro" as described in this same sentence.

- *Discussion – when describing effects neutrophils, Th17 and gd-T cells. Baek et al, Nat Com, 2017 should also be cited, as this paper directly describes influence of cholesterol/cholesterol metabolites on these cell types in the context of breast cancer.*

Response

The reference is now included.

- Discussion and citations of papers describing the purported endogenous ligands for the RORs *should also be included*.

Response

Discussion and citations are now included in the second paragraph of the result description.

- *Figure 2e: “none-tnbc” should read “non-tnbc”*

Response

Corrected.

- *Rather than the pictures of tumors/mice in Fig 6d and e, the authors should consider including the growth curves in Supplemental Fig 6c,j and K etc. It is much more impressive that the drug combinations work across several models.*

Response

As suggested, we now moved Supplementary Fig 6j and K to Figure 6.

Reviewer #4 (Remarks to the Author):

The study by Cai et al. identifies the nuclear receptor RORg as a novel master regulator of cholesterol metabolism and homeostasis in triple negative breast cancer. The authors go on to show that this protein is targetable with small molecule antagonists that selectively inhibit growth of human TNBC cells in vivo. This is a highly novel finding that has wide ramifications, including beyond breast cancer. Notably, the authors demonstrate that in TNBC cells, RORg is dominant over SREBP-2, a transcription factor well established as a master regulator of cholesterol metabolism. The findings identify important biochemical distinctions and dependences between TNBC and ER+ breast cancer. As a result, the study will have wide interest in the breast cancer community. However, the principal observation may be more fundamental and may have wide applicability in other cancers and other physiologic settings, likely generating interest much more widely. The study is comprehensively done, with multiple models, a wide range of experimental approaches, and an extensive data set in support of the conclusions of the study. Statistics appear satisfactory. There are some issues that could be addressed to strengthen the study further:

We very much appreciate this reviewer’s remarks on our study and his/her indication of our study being highly novel with wide ramifications including beyond breast cancer.

Major points:

1. Although human data are included, the treatment of human breast cancer data is rather superficial. RNA expression level of transcription factors is not a satisfactory indicator of activity. The RNA profiling and ChIP-seq data allow the development of an ROR γ activity signature that could be read out in human breast cancer RNA expression profiles and many other datasets. Because extensive human breast cancer datasets exist in the public domain, not having these data limits the translational impact of the study.

Response

We totally agree with this reviewer's comment. We have done the suggested analysis using data from RNA-seq and ChIP-seq and developed a ROR γ activity signature as shown below. Interestingly, all of the 14 ROR γ activity signature genes are actually involved in cholesterol biosynthesis pathway. Indeed, this signature can have a prognostic value in predicting the survival of TNBC patients as shown below. Since this data is part of our ongoing projects focusing on the clinical relevance of ROR γ expression and the expression of its target genes/proteins in breast cancers and other types of cancer, we feel it's more appropriate to present the data in those separate studies. Also, we have hard time finding the right place in this manuscript to include such data.

[Redacted]

2. The exclusive focus on TNBC vs. ER+ cancer is appropriate, but because the master regulator discovery could potentially operate in many other scenarios, it would be helpful if there was some sense from data presented here that this mechanism operates elsewhere. This could be done by analyzing inferred RORg activity using a gene signature panel as mentioned above.

Response

As described above, we have generated the ROR γ activity signature and also shown that it has prognostic value in predicting the survival of TNBC patients. Since this data is part of our ongoing projects focusing on the clinical relevance of ROR γ expression and the expression of its target genes/proteins in breast cancers and other types of cancer, we feel it's more appropriate to present the data in those separate studies.

3. There should be a better description of the specificity, selectivity and potential limitations of the RORg antagonists, since non-selectivity would confound one of the principal conclusions of the study.

Response

We agree that better description of the selection of ROR γ antagonists is needed. We now included additional descriptions of the compounds used as shown in the Results of page 6.

4. There is quite a bit of sloppiness in the text, particularly with respect to standard English.

Response

We apologize for the inconvenience. We now have performed a complete checking of English grammar for this manuscript.

Minor point:

The header label in figure 3e is mis-aligned.

Response

Corrected

References

- 1 Xiao, S. *et al.* Small-molecule ROR γ antagonists inhibit T helper 17 cell transcriptional network by divergent mechanisms. *Immunity* **40**, 477-489, doi:10.1016/j.immuni.2014.04.004 (2014).
- 2 Skogerson, K., Wohlgemuth, G., Barupal, D. K. & Fiehn, O. The volatile compound BinBase mass spectral database. *BMC Bioinformatics* **12**, 321, doi:10.1186/1471-2105-12-321 (2011).
- 3 Kind, T. *et al.* FiehnLib: mass spectral and retention index libraries for metabolomics based on quadrupole and time-of-flight gas chromatography/mass spectrometry. *Anal Chem* **81**, 10038-10048, doi:10.1021/ac9019522 (2009).
- 4 Zhu, S., Waguespack, M., Barker, S. A. & Li, S. Doxorubicin directs the accumulation of interleukin-12 induced IFN gamma into tumors for enhancing STAT1 dependent antitumor effect. *Clin Cancer Res* **13**, 4252-4260, doi:10.1158/1078-0432.CCR-06-2894 (2007).
- 5 ElBayoumi, T. A. & Torchilin, V. P. Tumor-targeted nanomedicines: enhanced antitumor efficacy in vivo of doxorubicin-loaded, long-circulating liposomes modified with cancer-specific monoclonal antibody. *Clin Cancer Res* **15**, 1973-1980, doi:10.1158/1078-0432.CCR-08-2392 (2009).
- 6 Mu, C. *et al.* Chemotherapy Sensitizes Therapy-Resistant Cells to Mild Hyperthermia by Suppressing Heat Shock Protein 27 Expression in Triple-Negative Breast Cancer. *Clin Cancer Res* **24**, 4900-4912, doi:10.1158/1078-0432.CCR-17-3872 (2018).
- 7 Santori, F. R. *et al.* Identification of natural ROR γ ligands that regulate the development of lymphoid cells. *Cell Metab* **21**, 286-298, doi:10.1016/j.cmet.2015.01.004 (2015).
- 8 Soroosh, P. *et al.* Oxysterols are agonist ligands of ROR γ and drive Th17 cell differentiation. *Proc. Natl. Acad. Sci. U. S. A.* **111**, 12163-12168, doi:10.1073/pnas.1322807111 (2014).
- 9 Hu, X. *et al.* Sterol metabolism controls T(H)17 differentiation by generating endogenous ROR γ agonists. *Nat Chem Biol* **11**, 141-147, doi:10.1038/nchembio.1714 (2015).
- 10 Schonewille, M. *et al.* Statins increase hepatic cholesterol synthesis and stimulate fecal cholesterol elimination in mice. *J Lipid Res* **57**, 1455-1464, doi:10.1194/jlr.M067488 (2016).
- 11 Mast, N., Bederman, I. R. & Pikuleva, I. A. Retinal Cholesterol Content Is Reduced in Simvastatin-Treated Mice Due to Inhibited Local Biosynthesis Albeit Increased Uptake of Serum Cholesterol. *Drug Metab Dispos* **46**, 1528-1537, doi:10.1124/dmd.118.083345 (2018).

Reviewers' Comments:

Reviewer #1:

Remarks to the Author:

In the revised manuscript, the authors have addressed my original concerns. However, some of the conclusions still need to be toned down.

In the abstract, the authors claim that "... that ROR γ functions as a master activator of the entire cholesterol-biosynthesis program,". Given the fact that ROR γ is only important in TNBC but not in ER+ cancer cells, the word "master" is an overclaim.

Related to my original point 1, if under overexpression condition, ROR γ can function independent of SREBP2 to induce the CB genes, it implies that ROR γ has SREBP2-independent mechanism in regulating the CB genes. This should be pointed out in the text.

Reviewer #2:

Remarks to the Author:

The authors answered my concerns correctly

Reviewer #3:

Remarks to the Author:

The authors have addressed this reviewer's concerns. I appreciate and am satisfied with their attempt to evaluate therapeutic efficacy against 4T1 tumors at 200mm³.

Reviewer #4:

Remarks to the Author:

The authors have done a comprehensive job of responding to the reviewer comments. I have no additional comments that require further review. This is a very important study with high impact. I appreciate that the authors want to withhold the bioinformatics data shown in the response to Reviewer 3, but I still think that would be a significant addition if it were included in this study.

Response to referees

Reviewer #1 (Remarks to the Author):

In the revised manuscript, the authors have addressed my original concerns. However, some of the conclusions still need to be toned down.

In the abstract, the authors claim that "... that ROR γ functions as a master activator of the entire cholesterol-biosynthesis program,". Given the fact that ROR is only important in TNBC but not in ER+ cancer cells, the word "master" is an overclaim.

We now removed "master" and changed the sentence in the abstract to "We demonstrate that ROR γ functions as an essential activator of the entire cholesterol-biosynthesis program,".

Related to my original point 1, if under overexpression condition, ROR γ can function independent of SREBP2 to induce the CB genes, it implies that ROR γ has SREBP2-independent mechanism in regulating the CB genes. This should be pointed out in the text.

We now pointed out that in the Results text as follows: "Interestingly, knockdown of SREBP2 had essentially no effect on the activation of the cholesterol-biosynthesis program by the ROR γ , suggesting a SREBP2-independent function of an overexpressed ROR γ (Fig. 3h)."

Reviewer #2 (Remarks to the Author):

The authors answered my concerns correctly.

We appreciate this reviewer's final comments.

Reviewer #3 (Remarks to the Author):

The authors have addressed this reviewer's concerns. I appreciate and am satisfied with their attempt to evaluate therapeutic efficacy against 4T1 tumors at 200mm³.

We appreciate this reviewer's final comments.

Reviewer #4 (Remarks to the Author):

The authors have done a comprehensive job of responding to the reviewer comments. I have no additional comments that require further review. This is a very important study with high impact. I appreciate that the authors want to withhold the bioinformatics data shown in the response to Reviewer 3, but I still think that would be a significant addition if it were included in this study.

We appreciate the understanding of this reviewer.